

# Ice-shelf freshwater triggers for the Filchner-Ronne Ice Shelf melt tipping point in a global ocean model

Matthew J. Hoffman[1], Carolyn Branecky Begeman[1], Xylar S. Asay-Davis[1], Darin Comeau[1], Alice Barthel[1], Stephen F. Price[1], and Jon Wolfe[1]

[1]Los Alamos National Laboratory

**Correspondence:** Matthew Hoffman (mhoffman@lanl.gov)

**Abstract.** Some ocean modeling studies have identified a potential tipping point from a low to high basal melt regime beneath the Filchner-Ronne Ice Shelf (FRIS), Antarctica, with significant implications for subsequent Antarctic Ice Sheet mass loss. To date, investigation of the climate drivers and impacts of this possible event have been limited, because ice-shelf cavities and ice-shelf melting are only now starting to be included in global climate models. Using a version of the Energy Exascale Earth System Model (E3SM) that represents both ocean circulations and melting within ice-shelf cavities, we explore freshwater triggers of a transition to a high melt regime at FRIS in a low resolution (30 km in the Southern Ocean) global ocean-sea ice model. We find that a realistic spatial distribution of iceberg melt fluxes is necessary to prevent the FRIS melt regime change from unrealistically occurring under historical reanalysis-based atmospheric forcing. Further, improvement of the default parameterization for mesoscale eddy mixing significantly reduces a large regional fresh bias and weak Antarctic Slope Front structure, both of which precondition the model to melt regime change. Using two different stable model configurations, we explore the sensitivity of FRIS melt regime change to regional ice-sheet freshwater fluxes. Through a series of sensitivity experiments prescribing incrementally increasing melt rates from the smaller, neighboring ice shelves in the eastern Weddell Sea, we demonstrate the potential for an ice-shelf melt "domino effect" should the upstream ice shelves experience increased melt rates. The experiments also reveal that modest ice-shelf melt biases in a model, especially at coarse ocean resolution where narrow continental shelf dynamics are not well resolved, can lead to unrealistic melt regime change downstream, and these ice-shelf melt teleconnections are sensitive to baseline model conditions. Our results highlight both the potential and the peril of simulating prognostic Antarctic ice-shelf melt rates in a low-resolution, global model.

## 1 Introduction

Mass loss from the Antarctic Ice Sheet is the greatest uncertainty in projections of future sea level rise due to the potential for destabilization of marine-terminating sectors of the ice sheet (Church et al., 2013). Ice shelves restrain the flow of the grounded ice behind them, and thinning of ice shelves due to intensified melting from the ocean below leads to flow acceleration and increased mass loss of grounded ice (Joughin et al., 2012; Gudmundsson, 2013; Reese et al., 2018; Gudmundsson et al., 2019; Zhang et al., 2020). Thus, changes in ocean conditions within the cavities beneath ice shelves can strongly control ice-sheet evolution.



Ice-shelf cavities around Antarctica can be classified as "cold" or "warm" based on the absence or presence of modified Circumpolar Deep Water, resulting in ice shelves with low ($\mathcal{O}(1 \text{ m yr}^{-1})$) and high melt rates ($\mathcal{O}(10 \text{ m yr}^{-1})$), respectively (Jacobs et al., 1992; Dinniman et al., 2016; Jenkins et al., 2016). Cold cavities, such as below the Filchner-Ronne Ice Shelf (FRIS), may transition to warm conditions through the intrusion of modified Circumpolar Deep Water, or modified Weddell Deep Water (mWDW) in the Weddell Sea (Hellmer et al., 2012). For FRIS, some modeling studies have shown the existence of

a tipping point from a stable, cold state to a stable, warm state when the intrusion of mWDW becomes amplified by invigorated overturning circulation (Hellmer et al., 2012; Thoma et al., 2015; Hellmer et al., 2017; Hazel and Stewart, 2020; Daae et al., 2020). Once triggered, the switch in regimes with respect to ocean cavity temperature and FRIS basal melt rates is rapid (occurring over one to two decades) and remains stable even after the removal of the perturbation that triggered it (Hellmer et al., 2017; Hazel and Stewart, 2020).

Under some greenhouse gas emissions scenarios, some climate models project that the FRIS tipping point may be crossed in the late 21st century (Hellmer et al., 2012; Timmermann and Hellmer, 2013; Hellmer et al., 2017), while in other models the tipping point is not reached regardless of emissions scenario (Naughten et al., 2018). Daae et al. (2020) showed that this tipping point can be reached through significant freshening of Dense Shelf Water (DSW) and shoaling of the thermocline at the continental slope. These conditions reduce the density contrast between the continental shelf and the open ocean, leading

to inflow of mWDW that was otherwise blocked by DSW (Daae et al., 2020; Hellmer et al., 2017). At the same time, recent studies have demonstrated that ocean properties and ice-shelf melt rates can be affected by remote conditions via the Antarctic Slope Current (Nakayama et al., 2014; Gille et al., 2016; Nakayama et al., 2020; Bull et al., 2021; Dawson et al., 2023). As such, ice-shelf basal meltwater can have far-reaching impacts as it is advected along the coast by the Antarctic Coastal Countercurrent, modifying DSW properties, affecting mWDW access to the continental shelf, and impacting Antarctic Bottom

Water production (Nakayama et al., 2014; Silvano et al., 2018; Nakayama et al., 2020).

    To date, the ocean models that have been used to understand the mechanisms affecting Antarctic ice-shelf basal melt, including the FRIS tipping point, have largely been regional in extent and/or are uncoupled from atmosphere models. In contrast, Earth system models are more useful for future projections but generally have coarse resolution and more simplified parameterizations of physical processes. The Energy Exascale Earth System Model (E3SM) is one of the first Earth system

models to include ice-shelf cavities and prognostic melt fluxes (Jeong et al., 2020; Comeau et al., 2022).

    Here, we present the results of E3SM ocean and sea-ice simulations at low resolution driven by historical atmospheric reanalyses, focusing on FRIS tipping point behavior and the model conditions leading to it. We find that the typical treatment of Antarctic freshwater fluxes in climate models, a distribution that is uniform along the Antarctic coast and confined $\mathcal{O}(100$ km) from the coast, quickly leads E3SM to cross the FRIS melt tipping point. We attribute this in part to the iceberg melt

term term; switching to an iceberg melt climatology dataset avoids the tipping point and allows E3SM to model FRIS subshelf circulations and melting well. However, a strong regional fresh bias and weak Antarctic Slope Front (ASF) remain, which may precondition the model to prematurely reach the tipping point under future forcing. Modifying the default mesoscale eddy mixing parameterization significantly reduces these biases, but fails to eliminate excessive melting of smaller ice shelves in the eastern Weddell Sea where the continental shelf is narrow. Motivated by these elevated proximate ice-shelf melt fluxes and the



sensitivity of the modeled FRIS melt regime to freshwater flux, we conduct a series of experiments wherein eastern Weddell
Sea ice shelves melt at above-observed rates. We find that melt fluxes representative of a partial transition from cold to warm
cavity conditions in this adjacent region are sufficient to trigger the FRIS melt regime change, with the threshold being sensitive
to the baseline model state. We discuss some challenges these ice-shelf melt teleconnections create in a low resolution global
model and the extent to which the model results suggest the potential for a real world ice-shelf melt "domino effect".

## 2   Methods


E3SM (Leung et al., 2020; Golaz et al., 2019) is an Earth system model with coupled model components for the atmosphere
(Rasch et al., 2019), ocean (Petersen et al., 2019), sea ice (Turner et al., 2021), land (Bisht et al., 2018), river (Li et al., 2015),
and ice sheets (Hoffman et al., 2018). Notable aspects of E3SM are the ability for all components to use variable resolution
meshes and formulation to run on advanced supercomputing architectures with an ultimate goal of exascale computing per-
formance (Leung et al., 2020). E3SM v1 simulations were organized into three science simulation campaigns, Water Cycle
(v1.0, Golaz et al., 2019), Biogeochemistry (v1.1, Burrows et al., 2020), and Cryosphere (v1.2, Comeau et al., 2022), the last of
which is used here. The E3SM v1.2 Cryosphere configuration introduced two notable capabilities for realistically representing
freshwater fluxes from Antarctica into the ocean (Comeau et al., 2022). The first is the addition of Antarctic ice-shelf cavities
and prognostic ice-shelf basal melt fluxes in the ocean model. The second is the representation of iceberg melt fluxes through
a prescribed monthly climatology from the reanalysis of Merino et al. (2016), instead of being applied uniformly around the
Antarctic coast following the CORE-II-IAF protocol (Large and Yeager, 2008), as had been done previously.

### 2.1   Ocean and sea ice model description

The ocean and sea ice components of E3SM are the Model for Prediction Across Scales-Ocean (MPAS-Ocean) (Petersen
et al., 2019) and Model for Prediction Across Scales-Sea Ice (MPAS-Seaice) (Turner et al., 2021). The models are built on a
common framework (as is the ice sheet model MPAS-Albany Land Ice; Hoffman et al., 2018) that defines spherical or planar
centroidal Voronoi meshes (Ringler et al., 2013), commonly used geophysical operators, input/output libraries, and paralleliza-
tion methods (Message Passing Interface and openMP). MPAS-Ocean uses a finite volume discretization on a staggered C-grid
of a three-dimensional, hydrostatic, Boussinesq approximation to the incompressible fluid flow equations. MPAS-Seaice uses
a combination of finite volume and finite element methods (Turner et al., 2021) to describe the flow of ice in a continuum ap-
proximation with elastic-viscous-plastic rheology (Hunke and Dukowicz, 1997) and complete thermodynamics (Turner et al.,
2013; Turner and Hunke, 2015). Both models use explicit forward time-stepping with subcycling of some fast processes.

Extensive details of MPAS-Ocean can be found in Petersen et al. (2019), Comeau et al. (2022) and associated references, but
some key aspects are summarized here. MPAS-Ocean employs a z* vertical coordinate (Adcroft and Campin, 2004; Petersen
et al., 2015) that is modified beneath ice shelves so that the top layer follows the ice draft and layer thicknesses are adjusted
to mitigate pressure-gradient errors (Comeau et al., 2022). The calculation of ice-shelf basal fluxes of mass, heat, and salinity
uses a standard parameterization of boundary layer turbulence (Hellmer and Olbers, 1989; Holland and Jenkins, 1999) with a





velocity-dependent transfer coefficient for heat and salt that is spatially uniform and calibrated to Antarctic-wide observations of ice-shelf basal melt rate (Rignot et al., 2013). For E3SM v1.2, ice-shelf cavities have a fixed geometry and do not evolve as melt occurs.

## 2.2 Baseline simulation configurations

The simulations presented here use the E3SM v1 low resolution ocean mesh (EC60to30 in Petersen et al., 2019) that features 60 km resolution at mid latitudes, refined to 30 km in equatorial and high latitude regions. Resolution in the Southern Ocean ranges from about 35 km at the coast to 50 km near the Subtropical Front. There are 60 vertical levels in the ocean mesh, ranging from 10 m thickness at the surface to 250 m at depth. The model uses a time step of 30 minutes. The E3SM v1.2 Cryosphere configuration is capable of being run both in a configuration with coupling between atmosphere, land, river, ocean and sea ice components (Comeau et al., 2022) and one with ocean and sea ice only. In this study we only present results from the ocean and sea ice configuration. This configuration uses the atmospheric forcing and prescribed terrestrial freshwater fluxes from the Coordinated Ocean-ice Reference Experiment Phase II with interannual forcing (CORE-II IAF) (Large and Yeager, 2008) together with sea-surface salinity restoring to a monthly climatology. As in other studies, the 62-year forcing cycle of 1948-2009 is repeated multiple times.

At low resolution, MPAS-Ocean uses the Gent-McWilliams parameterization (Gent and Mcwilliams, 1990) for the horizontal mixing induced by unresolved mesoscale eddies. The standard application in E3SM v1 uses a spatially and temporally constant bolus coefficient, which has a value of 600 $m^2$ $s^{-1}$ (Petersen et al., 2019) for simulations with prescribed atmospheric forcing. However, early E3SM v1 ocean simulations indicated that the use of a constant bolus coefficient value led to weak ocean circulation (Petersen et al., 2019). An alternative implementation was added to MPAS-Ocean, here referred to as "variable GM", that scales the bolus coefficient by the in situ stratification, resulting in a spatially- and temporally-variable value (Danabasoglu and Marshall, 2007; Comeau et al., 2022).

We present results from three model configurations of increasing sophistication (Table 1), all with active ice-shelf melt fluxes.

- **CGM-UIB**: The CGM-UIB run uses the Gent-McWilliams parameterization with a constant bolus coefficient and a uniform distribution of iceberg melt around the coast of Antarctica with a Gaussian smoothing spread over 300 km. The total magnitude of iceberg melt flux applied is 1187 Gt $yr^{-1}$, the approximate observed total calving flux for Antarctica (Rignot et al., 2013; Depoorter et al., 2013). This treatment of iceberg melt fluxes is equivalent to 45% of the Antarctic freshwater flux prescribed by CORE-II IAF. Spreading Antarctic freshwater fluxes around the coast is also traditionally how freshwater fluxes are represented in many climate models, including E3SM's default v1.0 configuration (Petersen et al., 2019; Golaz et al., 2019). This run was stopped at model year 100 after the FRIS melt regime tipping point was crossed in year 71.





**Table 1.** Table of baseline simulations conducted for this study. All simulations include prognostic ice-shelf melt fluxes and use the CORE-II IAF atmospheric forcing. The GM bolus coefficient is either constant or variable. Iceberg melt fluxes are either prescribed uniformly around the coast following the CORE-II protocol or represented by the Merino et al. (2016) climatology.

| Simulation | GM bolus | Iceberg melt flux | Years simulated | Year FRIS tipping point crossed |
|---|---|---|---|---|
| CGM-UIB | Constant | Uniform (CORE-II) | 1-100 | 71 |
| CGM-DIB | Constant | Data (Merino et al., 2016) | 1-210 | – |
| VGM-DIB | Variable | Data (Merino et al., 2016) | 1-210 | – |

- **CGM-DIB**: The CGM-DIB run uses the same Gent-McWilliams parameterization with a constant bolus coefficient and applies the Merino et al. (2016) data iceberg melt flux climatology. Thus, it is identical to CGM-UIB but with the application of a more realistic iceberg melt flux distribution.

- **VGM-DIB**: The VGM-DIB run uses the Gent-McWilliams parameterization with a spatially and temporally varying bolus coefficient and applies the Merino et al. (2016) data iceberg melt flux climatology. It is identical to CGM-DIB but with the improved treatment of the Gent-McWilliams parameterization discussed above.

We define FRIS melt regime change as an increase in mean FRIS basal melt rate exceeding 2 times its observed values sustained for the duration of the simulation. Since previous studies have identified this regime change as being associated with tipping points (Hellmer et al., 2017; Hazel and Stewart, 2020), we assume that tipping points have been reached by simulations that undergo a FRIS melt regime change.

For the CGM-DIB and VGM-DIB runs that avoided the FRIS melt regime tipping point, the oceanographic conditions in the third CORE-II cycle were similar to the second CORE-II cycle. This was considered sufficient spin-up, and these runs were stopped at model year 210. This end year was chosen as it allowed a complete 62-year cycle of forcing after year 140, which is used as a branch point for additional runs, as described in the following section.

### 2.3 Eastern Weddell prescribed melt branch simulations

As discussed below, our baseline simulations indicate that the FRIS melt regime and the associated eastern Weddell Sea continental shelf water mass properties are highly sensitive to land-ice freshwater flux. They also a reveal a significant high melt rate bias in the ice shelves northeast of FRIS, which is upstream of FRIS via the coastal current. To probe the potential sensitivity of modeled FRIS melt rates to this upstream ice-shelf melt bias in combination with ocean mean state biases, we conduct an ensemble of partially prescribed melt experiments that branch off of the CGM-DIB and VGM-DIB baseline simulations. In each branch run, we prescribe a spatially and temporally uniform melt rate for the ice shelves in the eastern Weddell Sea region, encompassing Brunt, Stancomb-Wills, Riiser-Larsen, Quar, and Ekström ice shelves (28° W to 10° W longitude). Ice-shelf basal melt fluxes are prognostic for all ice shelves, including FRIS, outside of this region. We perform branch runs from both





the CGM-DIB and VGM-DIB simulations to explore the relative sensitivity of those two model configurations to perturbations in ice-shelf melt fluxes.

The ensemble of sensitivity experiments is comprised of prescribed melt rates in the eastern Weddell Sea region of 0.58, 2, 3, 4, 8, and 16 m yr$^{-1}$. The low end value represents the mean melt rate for this region simulated by the CGM-DIB baseline run, slightly higher than the mean melt rate for the VGM-DIB baseline run of 0.46 m yr$^{-1}$. The high end value is comparable to the "warm shelf" conditions at Pine Island and Thwaites ice shelves, which have average melt rates of 14 and 27 m yr$^{-1}$, respectively (Adusumilli et al., 2020). We then progressively reduce the prescribed regional melt rate value by half for both CGM-DIB and VGM-DIB configurations until the FRIS melt regime change does not occur within a 62-year CORE-II cycle. The prescribed melt rates at values between those characteristic of warm and cold cavity conditions could be considered to approximate states with intermediate cross-shelf heat fluxes associated with mWDW. They also provide model insight into the real world potential for ice-shelf melt teleconnections. Where we prescribe melt fluxes, we set the associated latent heat flux to zero; because the ambient temperature and salinity beneath the ice shelves are generally incompatible with the imposed melt rates, extracting the associated latent heat fluxes from the ocean would exacerbate this inconsistency and generally causes large amounts of supercooling, which is undesired.

The prescribed-melt sensitivity experiments are branched from the baseline runs in year 141. This year is chosen because it is near the start of a period of increasing salinity on the eastern Weddell Sea continental shelf that lasts a number of decades (roughly years 135-190), which are conditions under which mWDW intrusions are less favorable. In other words, we select a time period in the CORE-II forcing cycle when the FRIS melt regime change is less likely to occur. This increases the likelihood that any regime changes that are simulated are primarily due to the prescribed melt fluxes and not the surface forcing. The branch runs are continued until either a FRIS melt regime change occurs or a complete forcing cycle is complete (62 years). Note that our choice of branch year means a restart of the CORE-II time-series occurs during the experiments, in year 187, which is 46 years after the perturbations are first applied.

## 3 Results

### 3.1 CGM-UIB

During the initial 70 years of the CGM-UIB simulation, the model reproduces the observed magnitude (Fig. 1a,b) and, to a lesser extent, the spatial distribution of Antarctic ice-shelf basal melting (Fig. 2a,b). However, after that, total Antarctic ice-shelf melt flux nearly triples due to a large and rapid increase in melting at FRIS (Fig. 1a,b). We first evaluate these simulation results prior to reaching the FRIS melt regime change, focusing on the Weddell Sea, followed by a description of the FRIS melt regime change in Sect. 3.1.2.



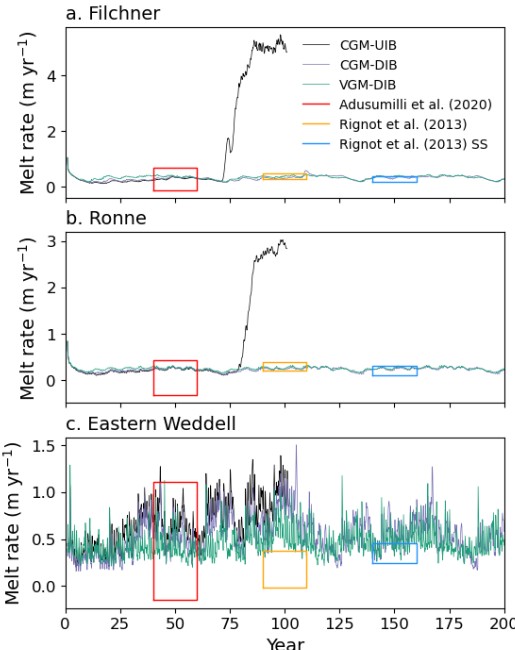

**Figure 1.** Time series of modeled melt rates averaged over (a) Filchner Ice Shelf, (b) Ronne Ice Shelf, and (c) Eastern Weddell ice shelves consisting of Brunt, Stancomb-Wills, Riiser-Larsen, Quar, and Ekström ice shelves. The mean ± standard deviation melt rates in each region are represented with boxes placed at arbitrary years for the following: a 2003-2008 satellite-derived estimate (Rignot et al., 2013), the melt rates required to maintain 2003-2008 steady state ice shelf extent (SS) (Rignot et al., 2013), and a 2010-2018 satellite-derived estimate (Adusumilli et al., 2020) .

### 3.1.1 Before FRIS melt regime change

After an initial adjustment period, the area-averaged melt rate for FRIS is within the range of observational uncertainty for both Filchner and Ronne ice shelves (Adusumilli et al. (2020); Rignot et al. (2013); Fig. 1a, b). FRIS is large enough to be reasonably well resolved at the horizontal resolution of 35 km. However, the modeled melt rate at smaller, nearby ice shelves in the eastern Weddell Sea is too high, by a factor of four or more (Fig. 1c). These ice shelves are poorly resolved at coarse resolution, as is the continental shelf, which is much narrower in this region relative to that for FRIS.

That FRIS is adequately resolved is evidenced by a generally good match of the spatial pattern of modeled basal melting to observations (Fig. 2a,b). Similar to observations, highest melt rates occurs near the grounding lines of tributary glaciers to the ice shelf, and freezing occurs in the central portion of both Ronne and Filchner ice Shelves. However, while the area-averaged melt rate matches observations, the local magnitude of both melting and refreezing is generally smaller than observed. A notable exception to the muted spatial variability in melt flux is that the magnitude of melting near the grounding lines of tributary glaciers is generally larger than observations.



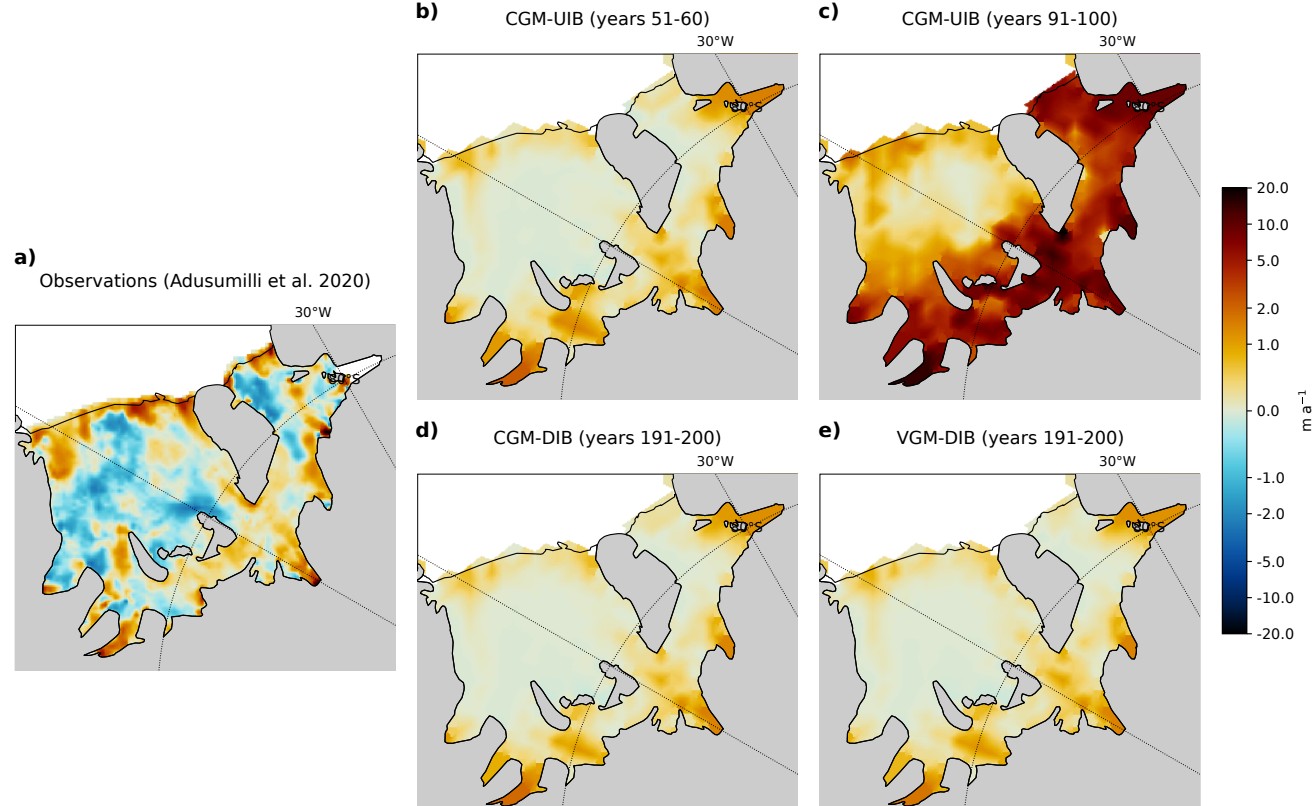

**Figure 2.** FRIS basal melt rates (m yr$^{-1}$ of freshwater). a) basal ice-shelf melt rate from Adusumilli et al. (2020). b) CGM-UIB run averaged over years 51-60. c) CGM-UIB run averaged over years 91-100 after the melt regime change has occurred. d) CGM-DIB run averaged over years 191-200. d) VGM-DIB run averaged over years 191-200. Note the nonlinear color scale.

Ocean circulation modeled beneath FRIS in the CGM-UIB simulation during the initial 70 years also follows some of the expected patterns (Nicholls et al., 2009; Hazel and Stewart, 2020; Daae et al., 2020), despite the relative low resolution of the model (Fig. 3a). There are two primary inflow points: the Ronne Depression along the west side of the Ronne Ice Shelf and near

Berkner Bank on the east side of Ronne Ice Shelf. A major outflow of dynamical importance for the tipping-point processes we observe is located in the Filchner Trough on the western side of the Filchner Ice Shelf, through which a combination of DSW and Ice Shelf Water (ISW) flows northward. While there are no extensive observations of velocity beneath FRIS, our modeled velocities are about five times smaller than those produced by other models (Daae et al., 2020; Bull et al., 2021). The weaker sub-ice-shelf circulation may be partly explained by our significantly lower horizontal resolution and the absence of

tides in our model. Additionally, Bull et al. (2021) showed that the strength of subshelf circulation is reduced as Weddell Sea continental shelf water is made fresher, a bias present in CGM-UIB.

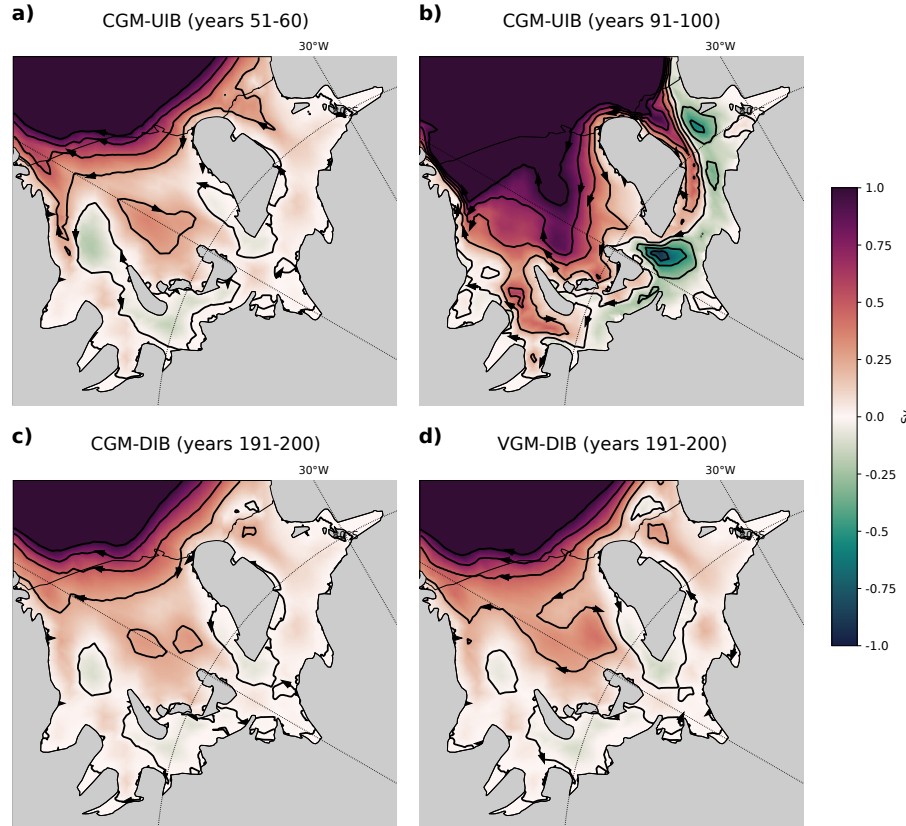

**Figure 3.** FRIS barotropic streamfunction (Sv) modeled by E3SM. Streamlines show direction of depth-integrated transport, spaced at 0.25 Sv intervals. a) CGM-UIB run averaged over years 51-60. b) CGM-UIB run averaged over years 91-100 after the melt regime change has occurred. c) CGM-DIB run averaged over years 191-200. d) VGM-DIB run averaged over years 191-200.

Water mass properties on the Weddell Sea in the CGM-UIB simulation have a strong fresh bias and a modest warm bias. Temperature and salinity are initialized from the Polar Science Center Hydrographic Climatology (PHC; Steele et al., 2001), and biases develop as the simulations progress. Fig. 4a shows temperature and salinity on the Weddell Sea continental shelf
averaged over years 51-60, prior to the regime change in the control run, relative to the water mass definitions in Naughten et al. (2018) (defined in the Fig. 4). In the CGM-UIB simulation, the primary water mass on the continental shelf is LSSW, the lighter variant of DSW. World Ocean Atlas 2018 observations (WOA18; Locarnini et al., 2019; Zweng et al., 2019) indicate that a significant volume of HSSW is present on the continental shelf (Fig. 4d), but HSSW is effectively absent from CGM-UIB by year 20. Furthermore, the higher density range of LSSW (>1027.7 kg m$^{-3}$) is absent from all runs by the end of the first
CORE-II cycle at year 62. Thus, DSW in our simulations is characterized by LSSW rather than a combination of LSSW and HSSW. The second largest water mass on the continental shelf is mWDW, but it and WDW occur in larger quantities than in the observations. ISW is present in the model in lower volume and at lower salinity than observations.





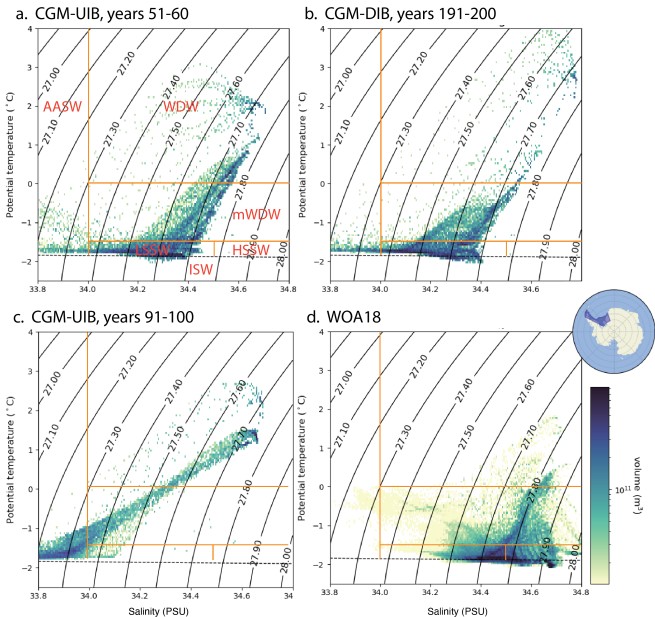

**Figure 4.** Temperature-salinity distribution for the western Weddell Sea continental shelf for depths shallower than 1000 m. a) CGM-UIB run averaged over years 51-60. b) CGM-DIB run averaged over years 191-200. c) CGM-UIB run averaged over years 91-100, after the melt regime change has occurred. d) Observations from World Ocean Atlas 2018. Boundaries for water masses are shown with orange lines and follow Naughten et al. (2018), renamed for Weddell Sea water masses, consistent with Kerr et al. (2018): Antarctic Surface Water (AASW), Weddell Deep Water (WDW), modified Weddell Deep Water (mWDW), Low Salinity Shelf Water (LSSW), High Salinity Shelf Water (HSSW), and Ice Shelf Water (ISW).

### 3.1.2 FRIS melt regime change

While the present day melt rates and circulation are represented reasonably well for the first few decades, the CGM-UIB
simulation exhibits an approximately tenfold increase in melt rate over a period of about ten years (Fig. 1). This melt regime changes in year 71 for Filchner Ice Shelf, followed six years later for Ronne Ice Shelf. Once the transition to the high melt regime occurs, melt rates remain elevated for the rest of the simulation. After the tipping point is passed, the entirety of Filchner Ice Shelf experiences highly elevated melt rates, as does the southern portion of Ronne Ice Shelf (Fig. 2c). Notably, the large scale circulation beneath FRIS reverses after the regime change, with strong inflow along the eastern side of Filchner Ice
Shelf extending clockwise around Berkner Island (Fig. 3b). Flow beneath Ronne Ice Shelf becomes less coherent, with high velocities around the ice rises in the southern portion. These changes in circulation are consistent with those reported by higher resolution models in the high-melt regime for FRIS (Hazel and Stewart, 2020; Daae et al., 2020).

The FRIS melt regime change is caused by the intrusion of mWDW onto the continental shelf and beneath the Filchner Ice Shelf via the Filchner Trough, as simulated in previous modeling studies (Hellmer et al., 2012, 2017; Daae et al., 2020;
Naughten et al., 2021). Prior to the melt regime change, the continental shelf seafloor is primarily occupied by cold DSW, with





small incursions of mWDW from the open ocean at the Filchner Sill and the Ronne Depression (Figs. 5a, 6a). It is the Filchner Trough pathway of mWDW that leads to the melt regime change when the warm mWDW intrusion extends beyond the ice shelf front for a sustained period. Following the intrusion, mWDW eventually fills the majority of the Filchner Ice Shelf cavity, wraps around Berkner Island, and displaces DSW throughout the majority of Ronne Ice Shelf (Figs. 5b, 6b). This can be seen
in the T-S plot for after the melt regime change (Fig. 4c), where there is a clear mixing line between WDW source water mixed with AASW and an almost complete absence of DSW.

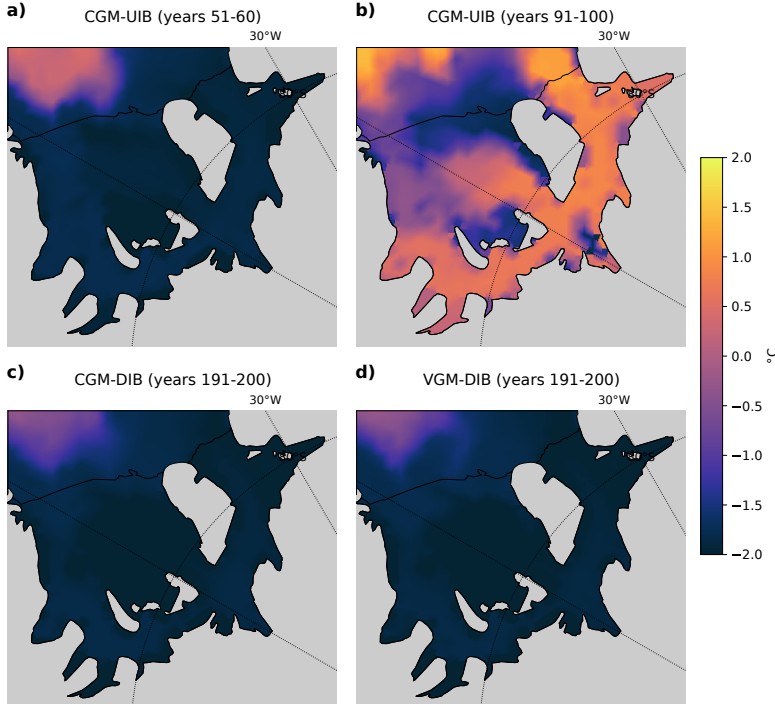

**Figure 5.** Seafloor potential temperature. a) CGM-UIB run averaged over years 51-60. b) CGM-UIB run averaged over years 91-100, after the melt regime change has occurred. c) CGM-DIB run averaged over years 191-200. d) VGM-DIB run averaged over years 191-200.

## 3.2 CGM-DIB

In contrast to the CGM-UIB simulation, the switch to a more realistic distribution of iceberg melting in the CGM-DIB simulation averts the FRIS melt regime change through the end of the 210 years simulated (Fig. 1a,b). The FRIS basal melt
distribution looks similar to the CGM-UIB simulation prior to the melt regime change and similar to observations (Fig. 2a-c). The subshelf circulation also is similar to the early part of the CGM-UIB run (Fig. 3a,c).

The change in iceberg freshwater flux distribution from closer to the coast in CGM-UIB to further from the coast in CGM-DIB results in an increase in the salinity of DSW on the Weddell continental shelf (Fig. 7). These differences are not apparent





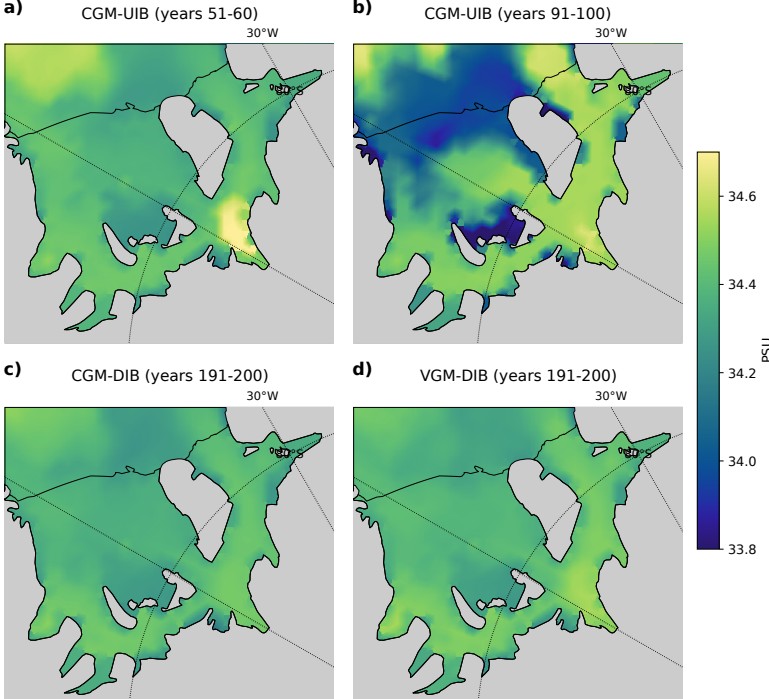

**Figure 6.** As in Fig. 5 but showing seafloor salinity.

in the temperature-salinity distribution shown in Fig. 4a,b but can be seen locally in the Filchner Trough (Fig. 8). While the
continental shelf salinity is still too low relative to observations, this water mass is sufficiently dense to limit mWDW intrusions
onto the shelf. This reduction in mWDW intrusions can be seen in Figure 8a, where mWDW intrusions only reach site M31W,
∼170 km from the continental shelf break (Ryan et al., 2017), once every several decades.

## 3.3 VGM-DIB

Although the CGM-DIB run averts the FRIS melt regime change in this historical simulation, there are several characteristics
that make this simulation more prone to FRIS melt regime change than the observed system. The low continental shelf salinities
lead to reduced DSW blocking of mWDW intrusions, and the high stratification in the region leads to a more baroclinic Weddell
Gyre and a weaker ASF that is also more prone to mWDW intrusions. In this section, we highlight how a different treatment
of eddy fluxes in the VGM-DIB simulation ameliorates these issues.

As with CGM-DIB, the VGM-DIB simulation avoids the FRIS melt regime change (Fig. 1a,b) and produces FRIS ice-
shelf melt patterns that capture the major features of the observations (Fig. 2a,e). FRIS subshelf circulation also looks similar
(Fig. 3c,d). Western Weddell Sea continental shelf temperature and salinity in the VGM-DIB run (not shown) look very similar





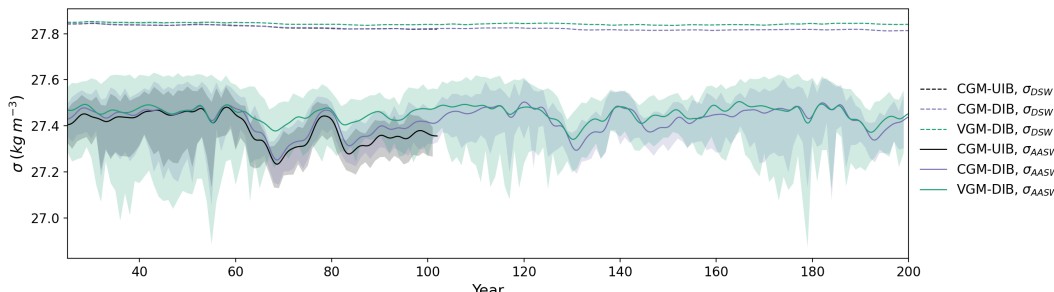

**Figure 7.** Potential density on the Weddell Sea continental shelf between $30°$W and $0°$W. $\sigma_{AASW}$ corresponds to the average monthly density above the thermocline (solid lines) and $\sigma_{DSW}$ corresponds to the average monthly density below the thermocline (dashed lines). Both quantities are low-pass-filtered in time with a cutoff of 3 months and the running annual minimum and maximum are bounded with shading. The thermocline depth is calculated as in Hattermann (2018).

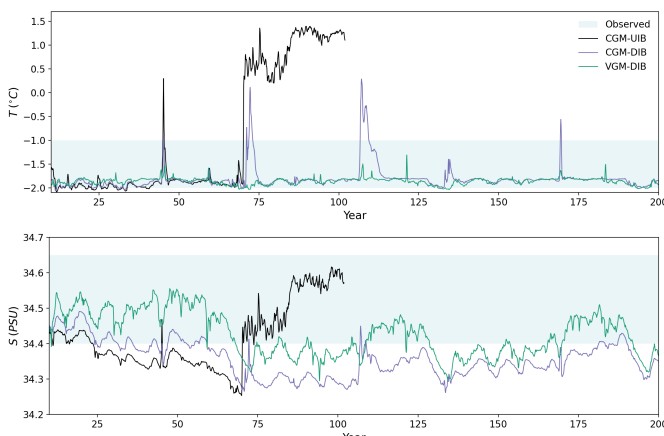

**Figure 8.** Time series of temperature (a) and salinity (b) at observational site M31W (Ryan et al., 2017) located in the Filchner Trough. The observational range is shaded.

to CGM-DIB (Fig. 4b). In addition, the periodic mWDW intrusions that reached M31W in both CGM runs are now absent (Fig. 8).

The ASF plays a critical role in modulating transport of heat onto the Weddell continental shelf. Our simulations consistently
feature weaker ASF characteristics than observations, making them more prone to mWDW intrusions. We characterize the ASF based on the thermocline depression from off-shelf to on-shelf; all our simulations show much smaller thermocline depressions than observed, roughly 200 m rather than 450 m (Hattermann, 2018). However, there is a notable improvement in the strength of the ASF in the VGM-DIB simulation (Fig. 9c), as seen in more steeply-sloping isopycnals at the continental shelf break and greater difference in depth of the thermocline between the shelf break and open ocean. The presence of a thinner warm layer
at the seafloor along the Filchner Trough in VGM-DIB relative to CGM-DIB (Fig. 9b,c) demonstrates the decreased access of



offshore mWDW to the continental shelf. The use of a spatially-varying bolus parameter in this run appears to have reduced cross-ASF heat transport and maintained a steeper ASF.

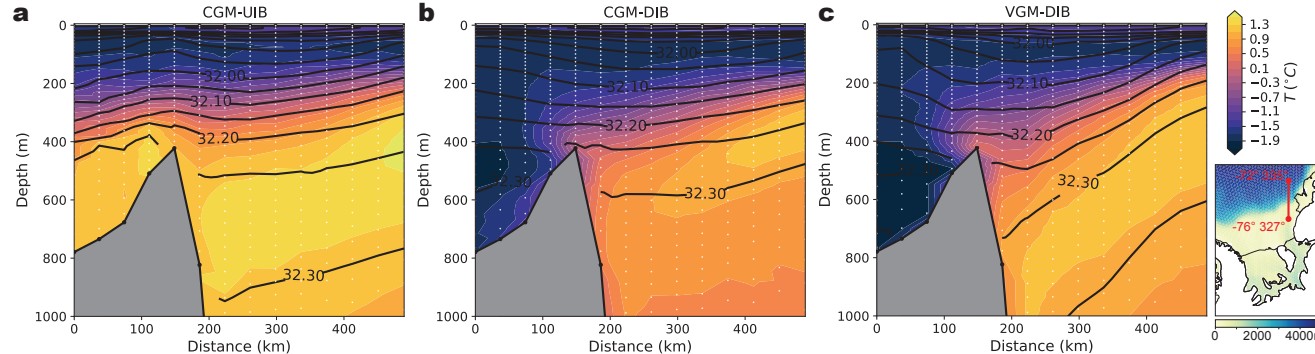

**Figure 9.** Cross sections of temperature (colors) and potential density anomalies (contours, referenced to 1000 db) along the Filchner Trough averaged over years 91-100 for CGM-UIB (a) and years 191 to 200 for CGM-DIB (b) and VGM-DIB (c). White dots indicate model data points used to construct the cross-section. [TODO: increase font size; replace single year averages with decadal averages; create cleaner inset map and move somewhere better; replace screengrab with tidied up version; add panel labels]

The other factor responsible for a more robust ASF in the VGM-DIB run is the reduction in salinity biases, improving the density structure on the continental shelf and at the shelf break. The fresh bias in the CGM-UIB and CGM-DIB simulations is
also associated with excessive near-surface stratification in the Weddell Gyre (Figure 9a,b). The result of this stratification is a more baroclinic Weddell Gyre, as less momentum from wind stress is transferred below the surface layer. Partially ameliorating these biases in the VGM-DIB run led to slightly less stratification (Figure 9c) and a more barotropic Weddell Gyre (Figure 10). The barotropic volume transport associated with the Weddell Gyre in VGM-DIB is within the 2-6 Sv range given by higher-resolution regional models with similar forcing and the Southern Ocean State Estimate (SOSE; Mazloff et al., 2010;
Wang et al., 2012) (Fig. 10). Barotropic transport in the Weddell Gyre is dynamically associated with a larger ASF thermocline depression through Ekman downwelling, which is consistent with our simulations (Hattermann, 2018).

Thus, the modified eddy representation in VGM-DIB affects mWDW transport both directly, through eddy fluxes across the ASF, and indirectly, through water mass changes. The relative importance of these direct and indirect effects is not possible to disentangle using our experimental design and existing model diagnostics. Regardless, the VGM-DIB simulation represents
the most realistic configuration of E3SM at this resolution for the Weddell Sea and FRIS.



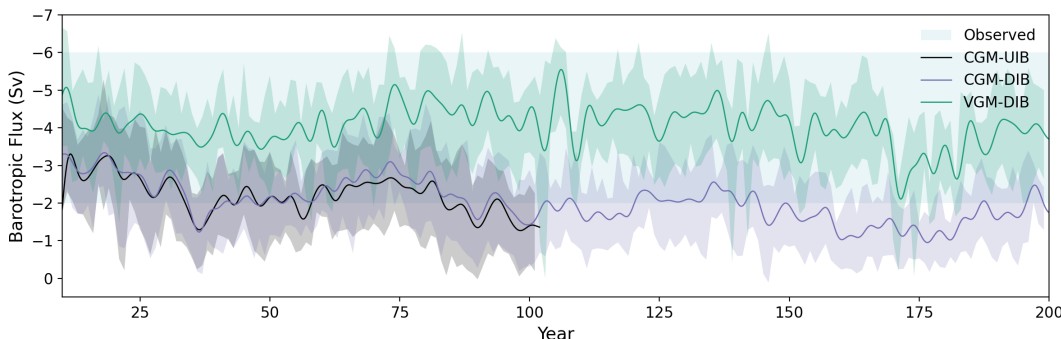

**Figure 10.** Barotropic transport in the Weddell Gyre for each simulation. The flux is computed from monthly-averaged velocities across Transect C in Wang et al. (2012) to facilitate comparison with reanalysis (75°S, 30°W to 72°S, 25°W). The barotropic fluxes are low-pass-filtered in time with a cutoff of 3 months and the running annual minimum and maximum are bounded with shading. Range from higher resolution models and reanalysis (Mazloff et al., 2010; Wang et al., 2012) shown with blue shading.



## 3.4 Prescribed-melt branch runs

As expected, the prescribed-melt branch runs with mean baseline melt fluxes for the eastern Weddell Sea ice shelves do not experience the FRIS regime change (Fig. 11a,b). However, modestly elevated melt rates applied to the eastern Weddell Sea ice shelves trigger the FRIS melt regime change. For CGM-DIB, all experiments with an eastern Weddell Sea melt rate of 2 m
$yr^{-1}$ and up lead to the FRIS regime change, whereas for VGM-DIB, the FRIS regime change occurs with eastern Weddell Sea melt rates of 4 m $yr^{-1}$ and up. As seen in the CGM-UIB baseline run, in all runs exhibiting the melt regime change, Filchner Ice Shelf transitions to a high melt regime first, followed by Ronne Ice Shelf within a decade. While the lowest melt rates applied do not lead to the occurrence of the FRIS melt regime change, all branch runs were stopped after 62 years (a complete CORE-II forcing cycle), and we are not able to rule out the possibility of these lead to a FRIS melt regime change later. All
simulations that lead to the FRIS regime change begin that transition within 21 years of the imposed eastern Weddell Sea melt rates. Increasing the melt rate applied reduces the time until the FRIS melt regime change occurs (Fig. 11c). This effect is sensitive to model baseline state, with CGM-DIB reaching the tipping point more quickly and at lower prescribed melt rates than VGM-DIB, consistent with the behavior in the previous simulations with fully prognostic ice-shelf melt fluxes.

The simulations that are slower to initiate transition to the elevated FRIS melt regime experience a longer transition period
(Fig. 11a,b). They also exhibit a non-monotonic increase in Filchner Ice Shelf melt rate during the transition (Fig. 11a). These temporary increases in Filchner Ice Shelf melt rate appear to be from pulses of mWDW intrusion (similar to those shown in Fig. 8) driven by surface forcing from which melt rates partially recover before the next surface forcing event occurs. For the simulations experimenting rapid transition to high Filchner Ice Shelf melt rates, the reorganization of the subshelf circulation is reinforced too quickly for these variations in surface driven mWDW intrusion to be exhibited.

Notably, the magnitude of FRIS melt rates after the regime change is a function of both the applied melt perturbations in the eastern Weddell Sea and the eddy parameterization (Fig. 11a,b). Higher imposed melt rates in the eastern Weddell Sea lead to larger post-transition FRIS melt rates. The CGM-DIB configurations yield higher post-transition FRIS melt rates than VGM-DIB for both Filchner and Ronne ice shelves.



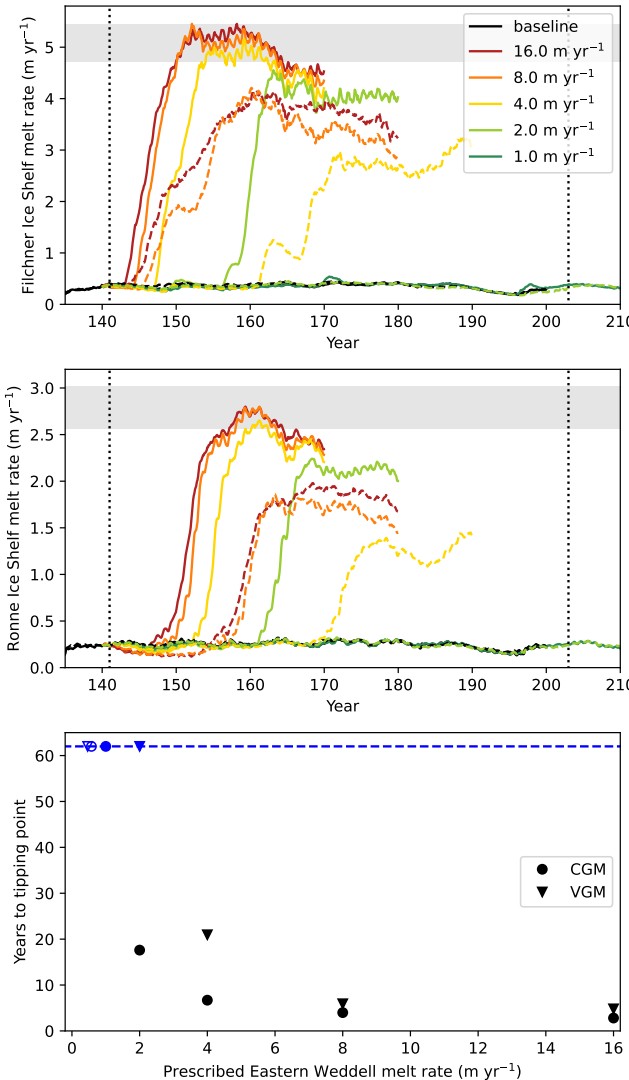

**Figure 11.** Results of branch runs prescribing melt rates at eastern Weddell Sea ice shelves. All branch runs were begun in year 141 and stopped after 62 years (one complete CORE-II cycle), indicated by vertical dotted lines. a) Area-averaged melt rate modeled at Filchner Ice Shelf. CGM-DIB is shown with solid lines, and VGM-DIB is shown with dashed lines. The baseline simulations are shown in black. The gray shading indicates the range of melt rates simulated after the regime change in the CGM-UIB run. b) Area-averaged melt rate modeled at Ronne Ice Shelf. Line styles same as for panel a. c) Number of years to occurrence of FRIS tipping point as a function of the melt rate prescribed at the eastern Weddell Sea ice shelves for both CGM-DIB (circles) and VGM-DIB (triangles). The open symbols represent the baseline runs. The initiation of the FRIS tipping point is defined here as the first year in which the modeled Filchner Ice Shelf melt rate exceeds twice the mean baseline value. The blue symbols on the blue dashed line indicate simulations that did not reach the FRIS tipping point within a complete CORE-II forcing cycle.



## 4 Discussion

### 4.1 Mechanisms for FRIS melt regime change

The fact that the CGM-UIB simulation undergoes a FRIS melt regime change under historical forcing is clearly inconsistent with historically observed low FRIS melt rates. Of the possible explanations for this inconsistency, two plausible candidates are that 1) the simulated state is closer to the FRIS tipping point than the historical state and that 2) the model's process representations shift the tipping point in state space relative to the real-world tipping point. We discuss these two candidates in the context of our suite of simulations, which feature both changes in the simulated state and in the model's representation of eddy processes. First, we show that our simulated state in CGM-UIB is consistent with the two ocean conditions for the FRIS regime change identified by Daae et al. (2020) using systematic perturbations to a high-resolution coupled ocean and sea ice model. Then we examine each condition in relation to changes in model representations across our simulation suite: the switch from spatially uniform to variable iceberg fluxes and from constant to variable GM eddy parameter.

Daae et al. (2020) demonstrated in their modeling study that both low continental shelf salinity and thermocline shoaling at the continental shelf break were necessary for a FRIS regime shift. The CGM-UIB simulation meets both of these criteria for FRIS regime shift (Figs. 4a,c, Fig. 9a) and manifests that regime shift. Weddell Shelf salinities in CGM-UIB are much lower than observed (Fig. 4a), consistent with the "strong freshening" scenario of Daae et al. (2020) in which DSW salinities are less than 34.4 PSU (the maximum salinity of DSW in CGM-UIB is 34.3 PSU). The CGM-UIB simulation also has a thermocline that is between 200 m and 300 m shallower than observed (not shown). In the simulations of Daae et al. (2020), thermocline shoaling of 200 m was sufficient to cause the FRIS regime shift. Thus, the lower resolution CGM-UIB simulation is at least as sensitive to the FRIS melt regime change as the higher resolution simulations of Daae et al. (2020).

The DSW salinity bias in our simulations is likely due to a combination of factors. One factor examined in Comeau et al. (2022) is the representation of mesoscale eddy fluxes; the VGM representation of mesoscale eddy fluxes results in much higher continental shelf salinities. In CGM-UIB, there is an additional contribution to DSW freshening by preferential iceberg fluxes near the coast. These iceberg fluxes freshen the surface more than at depth, enhancing the stratification between AASW and DSW (Fig. 9a). This stratification further contributes to DSW freshening because it inhibits the wintertime convection that would normally restore DSW salinities. Similarly, the addition of ice-shelf melt fluxes has been shown to increase stratification in E3SM and exacerbate DSW salinity biases on the Weddell continental shelf (Jeong et al., 2020).

The change in iceberg flux distribution between CGM-UIB and CGM-DIB has small effects on continental shelf density (Fig. 7), but is sufficient to avert the regime shift. This suggests either a high sensitivity to DSW density or a sensitivity to cross-slope gradients in buoyancy fluxes. An increase in the gradient of buoyancy fluxes (decreasing from onshore to offshore) may weaken the ASF through enhanced baroclinic eddy formation associated with the frontal instability (Marshall and Radko, 2003). This process tends to flatten the ASF isopycnals and would reduce the barrier to mWDW intrusions. This buoyancy flux gradient effect is hypothesized to have contributed to a temporary increase in mWDW intrusion strength after a significant sea-ice melting event on the eastern Weddell continental shelf (Ryan et al., 2020). However, we did not find evidence for a significantly different ASF isopycnal slope between CGM-UIB and CGM-DIB (Fig. 9a,b), suggesting that these baroclinic





eddy fluxes, parameterized in our model, were not significantly different. Thus, we hypothesize that the FRIS regime shift in our simulations displays a high sensitivity to DSW salinities. The branch runs with different Eastern Weddell melt rates also

correspond to different diffuse DSW salinity perturbations in contrast to perturbations of the local buoyancy gradient. That these branch runs resulted in different timing of FRIS regime change also lends support to the hypothesis that DSW salinity is the more important regime change factor in our simulations. Thus, our study corroborates Timmermann and Hellmer (2013) and Daae et al. (2020) in finding that the continental shelf salinity is a major control on mWDW inflow.

The thermocline shoaling in our simulations is a consequence of modeled water mass biases in the region. The density

contrast between AASW and WDW is thought to exert a strong control on the thermocline depression (Hattermann, 2018). As this density stratification increases, the ASC flow becomes more baroclinic and the thermocline depression decreases (Hattermann, 2018; Daae et al., 2020). While the thermocline depression is lower than observations in all of our simulations (Fig. 9), the VGM parameterization of mesoscale eddy fluxes is sufficient to strengthen the Weddell Gyre and provide a dynamical barrier to WDW intrusions. Thus, VGM-DIB demonstrates how both a more realistic iceberg flux distribution and

a modification to the mesoscale eddy parameterization in coarse-resolution ocean models can prevent the regime shift at FRIS despite persistent water mass biases. Our results highlight the relevance of a two-pronged approach to improving the realism of FRIS regime change, by improving both ocean model state biases and process representation. In Section 4.3, we further discuss the prospects for accurate simulation of the FRIS regime change by climate models. In summary, we conclude that the propensity of this E3SM configuration to FRIS melt regime change is primarily due to biases in the simulated state, as opposed

to a shift of the tipping point in the model relative to the real-world; we find that our simulations trigger the melt regime change under similar physical conditions to other models, but that our model is more prone to the change because its biases place it unrealistically close to the tipping point.

## 4.2 Ice-shelf melt teleconnections in the model

Our model results provide clear evidence for the potential of ice-shelf basal melt teleconnections through the advection of

meltwater and its impact on continental shelf salinity and density. While this is the first study to our knowledge to explicitly link the melt flux between different ice shelves, this work builds on previous studies linking ice-shelf meltwater fluxes to distant ocean conditions and vice versa. In a high resolution ocean model, Nakayama et al. (2014) linked increasing ice-shelf melt in the Amundsen Sea to freshening in the Ross Sea via advection by the ASC, and Nakayama et al. (2020) identified that the freshening could extend to the Weddell Sea under large Amundsen Sea melt rates. The time scale of transport between adjacent

regions is a few years in our simulations, consistent with other higher resolution simulations (Dawson et al., 2023) and with the rapid initiation of FRIS melt regime change in our eastern Weddell Sea prescribed melt branch runs (Fig. 11). Nakayama et al. (2020) suggest that transport of freshwater anomalies is enhanced by strengthening of the ASC as density gradients across the ASF increase, an effect not investigated here and unlikely to be resolved well in our low resolution simulations.

Imposing freshwater fluxes representing Antarctic ice-shelf melt and similar "hosing" experiments have been conducted in

a number of global climate models that did not include prognostic ice-shelf basal melt fluxes (e.g. Fogwill et al., 2015; Phipps et al., 2016; Pauling et al., 2016, 2017; Bronselaer et al., 2018; Moorman et al., 2020). While these studies generally agree that





increased freshwater fluxes on the continental shelf lead to increased ocean stratification, the net effect on cross-shelf heat fluxes is nuanced. In low resolution models this stratification leads to increased warming at depth and an implied positive feedback in ice-shelf basal melting (Fogwill et al., 2015; Phipps et al., 2016; Bronselaer et al., 2018). In high resolution (eddy-permitting)

models, freshening on the continental shelf may weaken cross-shelf heat fluxes by strengthening density gradients across the ASF (Moorman et al., 2020), or enhance cross-shelf heat fluxes through baroclinic instability in the ASF and tidal mixing (Si et al., 2023). It is worth noting that the change in heat flux into the FRIS cavity may in fact be minor with a small degree of freshening ($\mathcal{O}(0.01)$ PSU) in the eastern Weddell ASC, as revealed by further regional eddy-permitting simulations (Bull et al., 2021). Thus, more work remains to reconcile the potential effects of freshwater fluxes and their spatial distribution on FRIS

regime change across the modeling studies that have been conducted thus far, particularly for larger freshwater perturbations such as explored here.

### 4.3 Challenges of representing Antarctic ice shelves in climate models

We have demonstrated the potential for a FRIS melt regime change in a CMIP-class ocean model due to biases in salinity and ASC strength in the Weddell Sea. While our model biases are larger than many regional ocean modeling studies, global

ocean models used in Earth system models do not have the benefit of regional lateral boundary conditions to constrain model behavior in the Weddell Sea. Given what we understand about the FRIS melt regime change, here we place our E3SM results in context and comment on the applicability of CMIP models in this region.

Barthel et al. (2020) evaluated ocean temperatures from 33 climate models in six Antarctic continental shelf regions against a 1979–2005 historical climatology of coastal water masses compiled from shipboard measurements, instrumented seals, and

reanalysis for selecting climate model forcing for the Ice Sheet Model Intercomparison Project for CMIP6 (ISMIP6). E3SM was not complete for CMIP5 so we show E3SM results in context of the CMIP5 models in Fig. 12. While the analysis shown by Barthel et al. (2020) only evaluated temperature, for E3SM simulations in Fig. 12 we show the 20-year rolling mean trajectory in T,S space. In addition to the ocean and sea ice simulations described in this study, we show the fully-coupled (atmosphere-land-river-ocean-sea ice) E3SM simulations described by Comeau et al. (2022). Note that because E3SM v1.2 was not applied

to historical simulations, the fully-coupled E3SM results shown in Fig. 12 represent constant preindustrial climate conditions, while the analysis of Barthel et al. (2020) considered the 1979–2005 historical period — the comparison is intended to be illustrative.

Overall, the E3SM simulations tend to be biased fresh and warm compared to observations. The fully-coupled CGM simulation described by Comeau et al. (2022) (thin purple line) exhibits an early drift to fresher biases. Over the course of its 160

years of simulation (by which time it exhibits a regime shift), it would be amongst the outliers in the CMIP5 ensemble (beyond 1.5 interquartile range from the mean, shown by the empty circles) with a temperature bias in this region that ranges from 0.8° (years 10-30) making it the worst of the CMIP5 ensemble, to 0.2° over years 79-99, exceeded by the five CMIP5 outliers. In contrast, the fully-coupled VGM simulation (green, see Comeau et al. (2022)) remains stable in (T,S) space, and within the temperature range covered by the CMIP5 models used for ISMIP6.





Of the ocean and sea ice historical simulations described in this study, the trajectory of CGM-UIB (thick black line) towards
the FRIS melt regime change can be seen by its rapid evolution to fresher, warmer conditions in the Weddell Sea, similar to the
fully-coupled CGM simulation (thin black line). The CGM-DIB simulation avoids the regime change, but still exhibits a warm
bias and freshening over the course of the simulation. Our favored configuration, VGM-DIB, also freshens over the simulation
but exhibits the smallest salinity and temperature biases. Overall, we note that despite its biases, E3SM is not outside of the
CMIP5 range, as it exhibits temperature biases ranging from middle-of-the-road CMIP models to the warmer CMIP5 outliers
depending on its parameterization. Issues such as a FRIS melt regime shift may be encountered by other models, particularly
once they account for freshwater from ice-shelf melt or icebergs.

While it is not possible to identify an absolute threshold in regional salinity that avoids a FRIS melt regime shift for all
models (Hellmer et al., 2017; Daae et al., 2020; Haid et al., 2022), this comparison for E3SM configurations with and without
regime change provide some guidance for climate model evaluation in this region. As recent studies have clearly identified the
importance of continental-shelf salinity for controlling FRIS melt regime through its impact on density (Daae et al., 2020; Bull
et al., 2021; Haid et al., 2022), we recommend that future evaluations of ocean models for forcing ice-sheet models consider
regional salinity in addition to temperature.

Our results demonstrate that, while the inclusion of ice-shelf cavities and prognostic ice-shelf basal melt rates are critical
for projecting changes in the Antarctic, the significant technical challenges of introducing these capabilities to Earth system
models are compounded by potential complications from regional model biases that may be difficult to improve in a global
climate model. While typical standalone parameterizations of ice-shelf basal melt (e.g. Jourdain et al., 2020) that are forced by
offshore ocean conditions would not produce a melt regime change even with a regional fresh bias, the same ocean conditions
can lead to melt regime change and rapid change to continental temperature and salinity with prognostic melt fluxes in an
ocean model. To date, the other CMIP-class ocean model with prognostic ice-shelf melt fluxes also exhibits a fresh bias in the
Weddell Sea (and Ross Sea), which leads to FRIS melt regime change in SSP5-8.5 projections around 2100 (Siahaan et al.,
2022). Typical features of CMIP class ocean models, such as modest resolution requiring parameterization of mesoscale eddies,
under-resolved coastal polynyas, usage of vertical mixing schemes not developed for polar conditions, simplified treatment of
iceberg freshwater fluxes, and lack of tides contribute to challenges in resolving the key processes of DSW formation and ASF
dynamics. Process studies and high-resolution modeling will remain critical for guiding improvement of low resolution global
ocean models applied in the Antarctic.



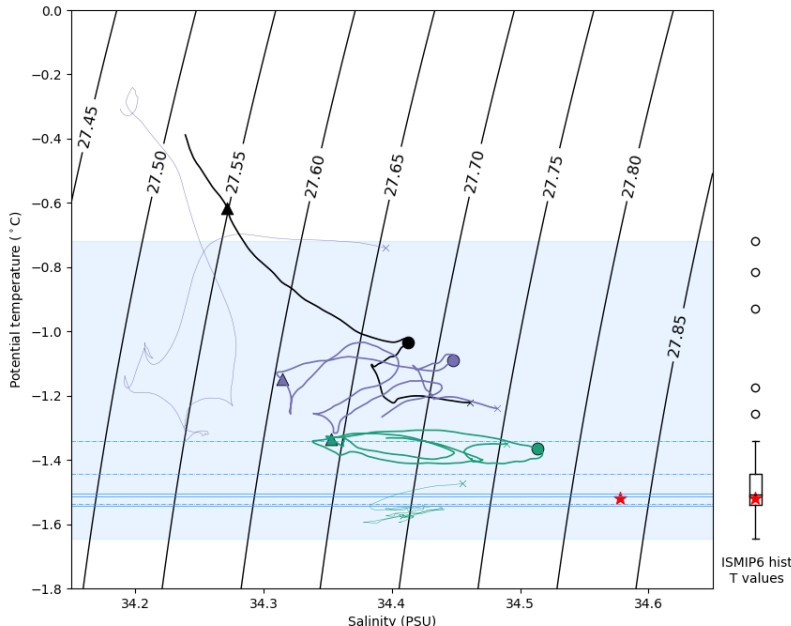

**Figure 12.** Comparison of modeled Weddell Sea continental shelf water mass properties in E3SM and CMIP5 models used in the ISMIP6 intercomparison. Observed temperature and salinity from WOA are indicated by the red star. The modeled temperature and salinity are represented by their 20-year-mean trajectories, with initial twenty-year average indicated by an x. The three primary simulations described in this paper are shown with thick lines and colored symbols (CGM-UIB: black; CGM-DIB: purple; VGM-DIB: green). Circles mark the average over years 42-62 (final twenty years before the end of first the CORE-II cycle) and triangles for the average over years 70-90 (final two complete decades before CGM-UIB begins regime shift). The equivalent fully-coupled E3SM preindustiral spin-ups from Comeau et al. (2022) are indicated by thin lines in corresponding colors (coupled CGM-DIB: purple; coupled VGM-DIB: green), covering 200 years. The temperature range from the ensemble of CMIP5 historical simulations evaluated by Barthel et al. (2020) is indicated by the blue shading, and also represented in more details by the box and whisker plot on the right-hand side (with outliers as circles). The CMIP5 models selected for ISMIP6 are added for context (top 3: blue lines, top 6: dashed blue lines).



## 5    Conclusions

We have investigated the occurrence of a FRIS basal melt regime change in low resolution E3SM v1.2 forced by historical atmospheric reanalysis. As seen in the fully coupled E3SM (Comeau et al., 2022), careful treatment of iceberg melt fluxes and the mesoscale eddy parameterization is necessary to achieve realistic simulations of this region that avoid a FRIS melt regime change. While moving the iceberg melt flux from uniform around the Antarctic coast to a realistic spatial distribution avoids the tipping point, switching to a spatially variable bolus coefficient in the Gent-McWilliams parameterization further improves continental shelf salinity, ASF structure, and barotropic transport in the region, despite lingering fresh surface biases leading to an overly stratified ocean and excess heat at depth. With these features, E3SM is able to produce the broad scale patterns of present day FRIS melt rates and cavity circulation even at relatively low horizontal ocean resolution.

To investigate sensitivity of FRIS to freshwater fluxes in the region, we conducted a series of perturbation experiments, where the ice shelves in the eastern Weddell Sea were given increasingly larger prescribed melt rates. We find that melt rates of 2-4 m yr$^{-1}$ are sufficient to trigger the FRIS melt regime change in E3SM, and the regime change initiates faster at higher upstream melt rates. This work explicitly identified the possibility of Antarctic ice-shelf melt teleconnections, building on previous work linking freshwater fluxes and ice-shelf melt rates around Antarctica. Because of the interplay between ice-shelf basal melt fluxes and ocean conditions that we find here, we caution against inferring ice-shelf melt rates from modeled ocean state without prognostic melt fluxes.

Finally, we put E3SM regional biases in context of other climate models that have been evaluated in the region and find that E3SM skill in this region is comparable to other CMIP models, and the improvements discussed in this paper improve model skill. We discuss challenges in adding prognostic basal melt fluxes to global climate models and highlight the importance in reproducing continental shelf salinity and ASF strength for simulating ice-shelf melting. Challenges remain in projecting ice-shelf melting in climate models due to their low resolution and lack of some key processes. Continuing to integrate knowledge gained from observations and process models is critical for projecting the state of the Southern Ocean under future climate and the associated impacts on the Antarctic Ice Sheet and sea-level change.

*Code and data availability.* The E3SM code is available at https://github.com/E3SM-Project/E3SM, and the model version used for the simulations presented here is E3SM v1.2 (doi 10.11578/E3SM/dc.20210309.1). Information about running the model is available at https://e3sm.org/model/running-e3sm. Simulation data used for this paper is available on ESGF at https://esgf-node.llnl.gov/projects/e3sm, listed under Cryosphere-v1.2 (CGM-DIB run listed as v1.2.1"CORE-IAF with ice shelf melt fluxes"; VGM-DIB run listed as v1.3 "CORE-IAF with ice shelf melt fluxes + 3DGM"). CGM-UIB run and prescribed-melt branch runs available upon request. Most of the analysis on the ocean component MPAS-Ocean was performed using MPAS-Analysis, available at https://github.com/MPAS-Dev/MPAS-Analysis (doi 10.5281/zenodo.4407459).



*Author contributions.* MJH conceptualized the research, and MJH, CBB, and XSAD developed the simulation plan methodology. DC, XSAD, JW, and MJH developed the software configuration of E3SM used in these simulations. DC and MJH conducted the simulations described here. CBB, MJH, XSAD, DC, and AB analyzed and visualized the simulation results. MJH and CB prepared the original manuscript draft, and all authors contributed to review and editing. SFP also contributed funding and resource acquisition.

*Competing interests.* The authors declare no competing interests.

*Acknowledgements.* This work was supported by the Biological and Environmental Research program, funded by the US Department of Energy (DOE), Office of Science. This work was also supported by the DOE Office of Science Early Career Research program. This research used a high-performance computing cluster provided by the BER Earth System Modeling program and operated by the Laboratory Computing Resource Center at Argonne National Laboratory. This research used resources of the National Energy Research Scientific Computing Center, a DOE Office of Science User Facility supported by the Office of Science of the U.S. Department of Energy under Contract No. DE-AC02-05CH11231.



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
