# Peer review of "Ice-shelf freshwater triggers for the Filchner-Ronne Ice Shelf melt tipping point in a global ocean-sea ice model"

_EGUsphere, 2023_

## Author Comment (AC1)

Ice-shelf freshwater triggers for the Filchner-Ronne Ice Shelf melt tipping point in a global ocean model

Author(s): Matthew J. Hoffman et al.

MS No.: egusphere-2023-2226

**Response to Reviewer 1**

*Reviewer comments are in black font, and our responses are in blue font.*

General comments:

This study investigates the tipping point of the Filchner-Ronne Ice Shelf (FRIS), based on a series of numerical experiments of a sea-ice and ocean model (including ice shelves) forced by present-day atmospheric boundary conditions. The sea ice-ocean model is a component of the Energy Exascale Earth System Model (E3SM) for wider climate sciences. In this study, the authors first show that by adjusting the iceberg meltwater distribution and the mesoscale eddy parameterization (GM coefficient), the unrealistic tipping event of the FRIS ice-shelf basal melt occurred in the standard configuration (CGM-UIB case) can be avoided in the CGM-DIB and VGM-DIB cases. Using the adjusted experimental setups, the authors further examine the increase in the FRIS melting in response to changes in upstream ice-shelf melting and show that a significant rise in EWIS ice-shelf melting can cause the FRIS to experience a tipping point. A rapid increase in ice-shelf basal melt substantially impacts inner ice-sheet changes, potentially contributing to the Antarctic ice-sheet mass loss and the subsequent global sea-level rise. In addition, while the substantial increase in ice-shelf basal melting due to the enhanced contribution of the warm Circumpolar Deep Water onto the southwestern Weddell continental shelf region has been reported in several modeling studies, it is still an interesting topic to be addressed in a coarse-resolution ocean model, which is a part of the climate model. For these reasons, I consider that the topic of this study falls within the scope of The Cryosphere. Although the manuscript in the present format may be clear for those familiar with E3SM and its ocean model biases, I found it very difficult for general readers (including me) to follow the logic smoothly. I suggest revising the manuscript to make it more understandable for a broader audience.

We appreciate the reviewer's feedback and suggestions for making the manuscript more accessible to a wider audience.

Major comments:

1. While the study highlighted the use of E3SM in many places (e.g., the abstract (L4-6), the last two paragraphs of the introduction (L51-64), and the methods section), actually, this study used only its ocean, sea-ice, and ice-shelf components of it with specifying atmospheric boundary conditions (CORE-II forcing for global sea ice-ocean models). The description throughout the manuscript should be careful not to give the impression that the experiments were conducted with the full ESM. This study is based on a global sea ice-ocean model, and I think it is sufficient to briefly describe these components are part of the E3SM in the methods section.

This is a fair suggestion, and we will improve the clarity by primarily referring to our model configuration as a "global ocean-sea ice model", while still providing the context of serving as the ocean and sea ice components of E3SM.

2. Throughout the manuscript, only a very limited area of the southern Weddell Sea (FRIS) is shown, and there is no information such as place names used in the manuscript. I think a map is required showing the broad area, including the Eastern Wedell Sea and major place names. In addition, it would be better to use the model representations of the coastline/ice-front line and grounding line (instead of the realistic coastal/grounding lines) to show the model's horizontal resolution clearly.

This is a good suggestion. We will add a figure with a location map.

3. In the experiments where the melt rate of the Eastern Weddell Sea is increased to 4-16m/yr (Section 2.3), I believe it is important to also express these rates in terms of volume (Gt/yr) per unit area. Considering observational data (Lauber et al. 2023), a melt rate increase to 4-16m/yr seems excessively high. If there are any justifications for this rate based on other modeling results or evidence, it should be included in the manuscript.

Lauber, J., Hattermann, T., de Steur, L., Darelius, E., Auger, M., Nøst, O. A., & Moholdt, G. (2023). Warming beneath an East Antarctic ice shelf due to increased subpolar westerlies and reduced sea ice. Nature Geoscience, 16(10), 877–885. https://doi.org/10.1038/s41561-023-01273-5

As described in the text, the high end for prescribed melt rates in the Eastern Weddell Sea is not meant to represent historical conditions or projections but a limiting value informed by the present-day highest mean ice-shelf melt rates in Antarctica (Pine Island Glacier). Our highest prescribed melt rates are also comparable to the modeled melt rates for these ice shelves from the high-end emissions scenario considered by Mathiot & Jourdain (2023) - an approximate 40-fold increase from the melt rates in the historical period. We will add this reference and comparison to our manuscript. Furthermore, the highest prescribed melt rates we selected are not particularly important, because we see the melt regime shift occur for melt rates 4-8x lower (mean melt rates of 2-4 m/yr).

4. I understand that freshwater supply and distribution can determine the FRIS tipping, but can you estimate the tipping conditions in ***this model*** by analyzing the freshwater balance on the continental shelf in front of the FRIS? For example, how much Gt or more of freshwater on the southwestern Weddell continental shelf would be required to cause the tipping point?

We appreciate the suggestion to characterize the conditions for FRIS tipping in our model in terms of the freshwater balance on the FRIS shelf. However, the focus of our manuscript is not to fully characterize these conditions, but rather to demonstrate that these conditions are contingent on modeling choices, such as the GM parameterization (e.g., Figure 11). We do show that our model's conditions are consistent with those identified by Daae et al. (2020), who performed a more comprehensive suite of simulations to characterize the conditions in their model configuration.

While it would be interesting to analyze the freshwater balance on the FRIS shelf, we do not have freshwater tracers in our simulation output to accurately characterize the contribution that upstream ice shelf melting or iceberg flux distribution makes to the FRIS continental shelf salinity. While we could compute the relative freshwater content using a reference salinity, this quantity reflects not only ice-shelf freshwater advection but also indirect effects mediated by sea ice and frontal dynamics. We do not believe any tipping conditions identified in this manner will be scientifically useful or practically useful in terms of identifying tipping conditions in other models or model configurations.

5. There are two very relevant new papers. The first one is missing, and the second one requires an update of the citation.

Mathiot, P., & Jourdain, N. C. (2023). Southern Ocean warming and Antarctic ice shelf melting in conditions plausible by late 23rd century in a high-end scenario. Ocean Science, 19(6), 1595–1615. https://doi.org/10.5194/os-19-1595-2023

Haid, V., Timmermann, R., Gürses, Ö., & Hellmer, H. H. (2023). On the drivers of regime shifts in the Antarctic marginal seas, exemplified by the Weddell Sea. Ocean Science, 19(6), 1529–1544. https://doi.org/10.5194/os-19-1529-2023

Thank you for pointing these out. We will incorporate the first one, which is clearly relevant, and update the reference for the second one now that it is published.

Specific comments:

6. L35-44: There are some useful literature (Thompson et al. 2018 for introducing Antarctic Slope Front/Current, Thoma et al. 2010 for remote linkage between Eastern Weddell Sea and Weddell Sea, and Kusahara and Hasumi 2013 for pathway of the EWIS meltwater). Please consider including the references.

Thompson, A. F., Stewart, A. L., Spence, P., & Heywood, K. J. (2018). The Antarctic Slope Current in a Changing Climate. Reviews of Geophysics, 56(4), 741–770. https://doi.org/10.1029/2018RG000624

Thoma, M., Grosfeld, K., Makinson, K., & Lange, M. A. (2010). Modelling the impact of ocean warming on melting and water masses of ice shelves in the Eastern Weddell Sea. Ocean Dynamics, 60, 480–489. https://doi.org/10.1007/s10236-010-0262-x

Kusahara, K., & Hasumi, H. (2013). Modeling Antarctic ice shelf responses to future climate changes and impacts on the ocean. Journal of Geophysical Research: Oceans, 118(5), 2454–2475. https://doi.org/10.1002/jgrc.20166

We will add these very appropriate references.

7. L73-74: This sentence is incorrect. Dinniman et al. (2016) summarized ocean models, which include ice-shelf cavities, and several ocean models with the same features have been developed since the publication.

The intended meaning here was "introduced into E3SM", i.e., relative to the previous version of E3SM. We will add that qualifier to the sentence to avoid ambiguity.

Dinniman, M., Asay-Davis, X., Galton-Fenzi, B., Holland, P., Jenkins, A., & Timmermann, R. (2016). Modeling Ice Shelf/Ocean Interaction in Antarctica: A Review. Oceanography, 29(4), 144–153. https://doi.org/10.5670/oceanog.2016.106

8. L104 Where did you restore the surface salinity, and with what intensity?

Sea surface salinity restoring is applied everywhere in the global ocean, except for beneath sea ice and inside ice-shelf cavities.  A piston velocity of 50 m/yr is used.  These details will be added to the manuscript.

9. L109-110: I think that a map of the variable GM calculated in the model is helpful to see the difference from the constant-value case.

It would be difficult to present the three-dimensional field of GM scaling in a meaningful way, but even if that were possible, it is not clear the 3d GM coefficients themselves would be more informative than the transects of the ASF in Fig. 9 that demonstrate the cumulative effect of the 3dGM implementation.  We will rewrite this sentence to make it clear that the 3d version of GM being discussed was also used in Comeau, et al. (2022) and is not a new feature being described for the first time.

10. L115-136: Maps of iceberg melt flux are also helpful to understand the differences among the experiments.

These maps are available in Comeau, et al. (2022), as well as in detail in the original source (Merino, et al., 2016), so we prefer to not add another figure with this information here.

11. Figure 1: Adding Gt/yr to the right axis also makes it easier to compare with other studies and between ice shelves.

This is a good suggestion, and we will add the additional axis.

12. Figure 2: Information of longitude and latitude is sparse, and it is very difficult to see the latitude information.

We plan to add a new figure with an overview map that will have more detailed location information. We will also adjust the placement of the latitude labels on this figure.

13. Figure 4: Is the small map for the T-S plot? If so, please enlarge the map and add depth contours.

That is a good suggestion to enlarge the map.  Depth contours are not necessary on this location map as we define the region based on a single depth contour, but we will be sure to include useful depth contours on the new location map figure we plan to include.

14. Figure 7: Without vertical profiles of water properties, it is very difficult to understand the figure.

We have given careful consideration to the question of what representations of water properties are needed to understand this figure and the dynamics in our model. We believe that the reader can get an adequate sense of the changes in stratification in our model from the combination of Figure 9 (transects of temperature with density contours) and Figure 7. We will strengthen the connections between these two figures in the text. The purpose of Figure 7 is to demonstrate the aggregate (spatially-averaged and temporally filtered) differences in water mass properties between our model configurations. Since the thermocline depth can vary in space and time, we first identify the thermocline depth and then spatially average properties above and below that depth. Providing a spatially-averaged vertical profile would obscure the thermocline, and we do not believe a particular space and time choice would be best suited for this figure. We chose to produce the transects in

Figure 9 at the location and decade that we believe is most representative of the conditions that lead to or avert the tipping point.

15. Figure 8: A map of the barotropic stream function is required, with pointing to the observation site M31W and transect C.

We will add site locations (M31W, transect C) on a site map.  We have maps of the barotropic stream function in Figure 3.

16. Figure 9: Could you add panels of vertical profiles of alongshore current to see the Antarctic Slope Current? Why did you use the potential density anomaly referenced to 1000 m? The other figures used the potential density referenced to the surface (Figs. 4, 7, and 12). Please remove TODO (I think this kind of error should be removed before submission, circulating the manuscript among the authors.).

Yes, we will add panels of vertical profiles of alongshore current to Fig. 9.  We referenced the potential density anomalies to 1000 m to make this figure directly comparable to Daae et al. (2020) Fig. 3, which used that reference.  But we can see the value to having all figures in this manuscript referenced to the same depth and would be willing to adjust this figure accordingly.  We apologize for the wayward TODO note - it was missed by all authors in our final circulation.

17. Figure 10: It is very difficult to see the observed range. Please consider revising the color.

We will add dashed lines to the observed range in order to provide clarity.

18. Figure 12: Please add the experiment names in the figure.

Good suggestion.  We will add labels with the experiment names.

Technical corrections:

19. Title: It is not clear to me what the word "ice-shelf freshwater" stands for in the title. "Antarctic coastal freshening" may be a candidate to replace the word.

The term "ice-shelf freshwater" encompasses the impacts of both iceberg melt distribution and the ice-shelf basal melting from the eastern Weddell Sea ice shelves that we investigate.  We appreciate that this may not be clear, but this was the best concise term we came up with.  We prefer it to the suggestion of "coastal freshening", as we believe that is potentially more ambiguous, as it could also encompass sea ice or precipitation driven freshening.  However, we will define what we mean by "ice-shelf freshwater" upon its first usage in the abstract on line 5.

---

## Author Response (AR1)

**Ice-shelf freshwater triggers for the Filchner-Ronne Ice Shelf melt tipping point in a global ocean model**

Author(s): Matthew J. Hoffman, et al.

MS No.: egusphere-2023-2226

**Summary of changes to manuscript in response to reviewer comments**

Reviewer comments are in black font. Our explanations of revisions are in purple font.

**Reviewer 1**

General comments:

This study investigates the tipping point of the Filchner-Ronne Ice Shelf (FRIS), based on a series of numerical experiments of a sea-ice and ocean model (including ice shelves) forced by present-day atmospheric boundary conditions. The sea ice-ocean model is a component of the Energy Exascale Earth System Model (E3SM) for wider climate sciences. In this study, the authors first show that by adjusting the iceberg meltwater distribution and the mesoscale eddy parameterization (GM coefficient), the unrealistic tipping event of the FRIS ice-shelf basal melt occurred in the standard configuration (CGM-UIB case) can be avoided in the CGM-DIB and VGM-DIB cases. Using the adjusted experimental setups, the authors further examine the increase in the FRIS melting in response to changes in upstream ice-shelf melting and show that a significant rise in EWIS ice-shelf melting can cause the FRIS to experience a tipping point. A rapid increase in ice-shelf basal melt substantially impacts inner ice-sheet changes, potentially contributing to the Antarctic ice-sheet mass loss and the subsequent global sea-level rise. In addition, while the substantial increase in ice-shelf basal melting due to the enhanced contribution of the warm Circumpolar Deep Water onto the southwestern Weddell continental shelf region has been reported in several modeling studies, it is still an interesting topic to be addressed in a coarse-resolution ocean model, which is a part of the climate model. For these reasons, I consider that the topic of this study falls within the scope of The Cryosphere. Although the manuscript in the present format may be clear for those familiar with E3SM and its ocean model biases, I found it very difficult for general readers (including me) to follow the logic smoothly. I suggest revising the manuscript to make it more understandable for a broader audience.

We appreciate the reviewer's feedback and suggestions for making the manuscript more accessible to a wider audience.

Major comments:

1. While the study highlighted the use of E3SM in many places (e.g., the abstract (L4-6), the last two paragraphs of the introduction (L51-64), and the methods section), actually, this study used only its ocean, sea-ice, and ice-shelf components of it with specifying atmospheric boundary conditions (CORE-II forcing for global sea ice-ocean models). The description throughout the manuscript should be careful not to give the impression that the experiments were conducted with the full ESM. This study is based on a global sea ice-ocean model, and I think it is sufficient to briefly describe these components are part of the E3SM in the methods section.

This is a fair suggestion, and we have improved the clarity by primarily referring to our model configuration as a "global ocean-sea ice model", while still providing the context of serving as the ocean and sea ice components of E3SM.  These adjustments have been made throughout the manuscript wherever E3SM is mentioned.  We have also updated the title to use the term "global ocean-sea ice model".

2. Throughout the manuscript, only a very limited area of the southern Weddell Sea (FRIS) is shown, and there is no information such as place names used in the manuscript. I think a map is required showing the broad area, including the Eastern Wedell Sea and major place names. In addition, it would be better to use the model representations of the coastline/ice-front line and grounding line (instead of the realistic coastal/grounding lines) to show the model's horizontal resolution clearly.

This is a good suggestion.  We have added a figure with a detailed location map and referenced it as appropriate.  We chose to make the figure use realistic feature representation for clarity, because the model resolution can be seen in other map figures.

3. In the experiments where the melt rate of the Eastern Weddell Sea is increased to 4-16m/yr (Section 2.3), I believe it is important to also express these rates in terms of volume (Gt/yr) per unit area. Considering observational data (Lauber et al. 2023), a melt rate increase to 4-16m/yr seems excessively high. If there are any justifications for this rate based on other modeling results or evidence, it should be included in the manuscript.

Lauber, J., Hattermann, T., de Steur, L., Darelius, E., Auger, M., Nøst, O. A., & Moholdt, G. (2023). Warming beneath an East Antarctic ice shelf due to increased subpolar westerlies and reduced sea ice. Nature Geoscience, 16(10), 877–885. https://doi.org/10.1038/s41561-023-01273-5

As described in the text, the high end for prescribed melt rates in the Eastern Weddell Sea is not meant to represent historical conditions or projections but a limiting value informed by the present-day highest mean ice-shelf melt rates in Antarctica (Pine Island Glacier).  Our highest prescribed melt rates are comparable to the modeled melt rates for these ice shelves from the high-end emissions scenario considered by Mathiot & Jourdain (2023) - an approximate 40-fold increase from the melt rates in the historical period.  We have added comparisons to Lauber et al. and Mathiot and Jourdain to the manuscript.  We have also listed the corresponding prescribed melt flux in Gt/yr for each melt rate value used.

4. I understand that freshwater supply and distribution can determine the FRIS tipping, but can you estimate the tipping conditions in ***this model*** by analyzing the freshwater balance on the continental shelf in front of the FRIS? For example, how much Gt or more of

freshwater on the southwestern Weddell continental shelf would be required to cause the tipping point?

We appreciate the suggestion to characterize the conditions for FRIS tipping in our model in terms of the freshwater balance on the FRIS shelf. However, the focus of our manuscript is not to fully characterize these conditions, but rather to demonstrate that these conditions are contingent on modeling choices, such as the GM parameterization (e.g., Figure 11). We do show that our model's conditions are consistent with those identified by Daae et al. (2020), who performed a more comprehensive suite of simulations to characterize the conditions in their model configuration.

While it would be interesting to analyze the freshwater balance on the FRIS shelf, we do not have freshwater tracers in our simulation output to accurately characterize the contribution that upstream ice shelf melting or iceberg flux distribution makes to the FRIS continental shelf salinity. While we could compute the relative freshwater content using a reference salinity, this quantity reflects not only ice-shelf freshwater advection but also indirect effects mediated by sea ice and frontal dynamics. We do not believe any tipping conditions identified in this manner will be scientifically useful or practically useful in terms of identifying tipping conditions in other models or model configurations.

5. There are two very relevant new papers. The first one is missing, and the second one requires an update of the citation.

Mathiot, P., & Jourdain, N. C. (2023). Southern Ocean warming and Antarctic ice shelf melting in conditions plausible by late 23rd century in a high-end scenario. Ocean Science, 19(6), 1595–1615. https://doi.org/10.5194/os-19-1595-2023

Haid, V., Timmermann, R., Gürses, Ö., & Hellmer, H. H. (2023). On the drivers of regime shifts in the Antarctic marginal seas, exemplified by the Weddell Sea. Ocean Science, 19(6), 1529–1544. https://doi.org/10.5194/os-19-1529-2023

Thank you for pointing these out.  We have incorporated these references and updated the Hais et al. reference.

Specific comments:

6. L35-44: There are some useful literature (Thompson et al. 2018 for introducing Antarctic Slope Front/Current, Thoma et al. 2010 for remote linkage between Eastern Weddell Sea and Weddell Sea, and Kusahara and Hasumi 2013 for pathway of the EWIS meltwater). Please consider including the references.

Thompson, A. F., Stewart, A. L., Spence, P., & Heywood, K. J. (2018). The Antarctic Slope Current in a Changing Climate. Reviews of Geophysics, 56(4), 741–770. https://doi.org/10.1029/2018RG000624

Thoma, M., Grosfeld, K., Makinson, K., & Lange, M. A. (2010). Modelling the impact of ocean warming on melting and water masses of ice shelves in the Eastern Weddell Sea.

Ocean Dynamics, 60, 480–489. https://doi.org/10.1007/s10236-010-0262-x

Kusahara, K., & Hasumi, H. (2013). Modeling Antarctic ice shelf responses to future climate changes and impacts on the ocean. Journal of Geophysical Research: Oceans, 118(5), 2454–2475. https://doi.org/10.1002/jgrc.20166

We have incorporated these very appropriate references.

7. L73-74: This sentence is incorrect. Dinniman et al. (2016) summarized ocean models, which include ice-shelf cavities, and several ocean models with the same features have been developed since the publication.

The intended meaning here was "introduced into E3SM", i.e., relative to the previous version of E3SM. We added this qualifier to the sentence to avoid ambiguity.

Dinniman, M., Asay-Davis, X., Galton-Fenzi, B., Holland, P., Jenkins, A., & Timmermann, R. (2016). Modeling Ice Shelf/Ocean Interaction in Antarctica: A Review. Oceanography, 29(4), 144–153. https://doi.org/10.5670/oceanog.2016.106

8. L104 Where did you restore the surface salinity, and with what intensity?

Sea surface salinity restoring is applied everywhere in the global ocean, with the exception of beneath sea ice and inside ice-shelf cavities. A piston velocity of 50 m/yr is used. These details have been added to the manuscript.

9. L109-110: I think that a map of the variable GM calculated in the model is helpful to see the difference from the constant-value case.

It would be difficult to present the three-dimensional field of GM scaling in a meaningful way, but even if that were possible, it is not clear the 3d GM coefficients themselves would be more informative than the transects of the ASF in Fig. 9 that demonstrate the cumulative effect of the 3dGM implementation. We have added a sentence referring to Comeau, et al. (2022) for details on the implementation and effects of adding variable GM to MPAS-Ocean.

10. L115-136: Maps of iceberg melt flux are also helpful to understand the differences among the experiments.

We have added inset maps to the new location map showing the maps of iceberg melt flux. Larger maps are available in Comeau, et al. (2022), as well as in detail in the original source (Merino, et al., 2016). We added an additional reference to the Comeau et al. figure that shows this.

11. Figure 1: Adding Gt/yr to the right axis also makes it easier to compare with other studies and between ice shelves.

This is a good suggestion, and we have added the additional axis.

12. Figure 2: Information of longitude and latitude is sparse, and it is very difficult to see the latitude information.

We have added a new figure with an overview map that will have more detailed location information.  We have also adjusted the placement of the latitude labels on this figure.

13. Figure 4: Is the small map for the T-S plot? If so, please enlarge the map and add depth contours.

We have enlarged the map and added contours on the new location map.

14. Figure 7: Without vertical profiles of water properties, it is very difficult to understand the figure.

We have given careful consideration to the question of what representations of water properties are needed to understand this figure and the dynamics in our model. We believe that the reader can get an adequate sense of the changes in stratification in our model from the combination of Figure 9 (transects of temperature with density contours) and Figure 7. To address this, we have strengthened the connections between these two figures in the text. The purpose of Figure 7 is to demonstrate the aggregate (spatially-averaged and temporally filtered) differences in water mass properties between our model configurations. Since the thermocline depth can vary in space and time, we first identify the thermocline depth and then spatially average properties above and below that depth. Providing a spatially-averaged vertical profile would obscure the thermocline, and we do not believe a particular space and time choice would be best suited for this figure. We chose to produce the transects in Figure 9 at the location and decade that we believe is most representative of the conditions that lead to or avert the tipping point.

15. Figure 8: A map of the barotropic stream function is required, with pointing to the observation site M31W and transect C.

We have added site locations on the new Figure 1 site map.  We have maps of the barotropic stream function in Figure 3.

16. Figure 9: Could you add panels of vertical profiles of alongshore current to see the Antarctic Slope Current? Why did you use the potential density anomaly referenced to 1000 m? The other figures used the potential density referenced to the surface (Figs. 4, 7, and 12). Please remove TODO (I think this kind of error should be removed before submission, circulating the manuscript among the authors.).

Yes, we have added panels of zonal velocity to this figure.  With the addition of velocity transects, we shifted the location of the transect to be normal to the shelf break along the east side of Filchner Trough.  We also changed the time period for the CGM-UIB panels to be the year 51-60 decade prior to the melt regime change so comparisons with the other simulations are more direct.

We had referenced the potential density anomalies to 1000 m to make this figure directly comparable to Daae et al. (2020) Fig. 3, which used that reference.  But we can see the value to having all figures in this manuscript referenced to the same depth and have adjusted this figure accordingly.

17. Figure 10: It is very difficult to see the observed range. Please consider revising the color.

We have added dashed lines to the observed range in order to provide clarity. We have also updated the contents of this figure due to an issue with an incorrect transect location we had used originally. The relative differences between the three model runs remain similar.

18. Figure 12: Please add the experiment names in the figure.

Good suggestion. We have added a legend with the experiment names.

Technical corrections:

19. Title: It is not clear to me what the word "ice-shelf freshwater" stands for in the title. "Antarctic coastal freshening" may be a candidate to replace the word.

The term "ice-shelf freshwater" encompasses the impacts of both iceberg melt distribution and the ice-shelf basal melting from the eastern Weddell Sea ice shelves that we investigate. We appreciate that this may not be clear, but this was the best concise term we came up with. We prefer it to the suggestion of "coastal freshening", as we believe that is potentially more ambiguous, as it could also encompass sea ice or precipitation driven freshening. However, to clarify the term in the title, we have defined what we mean by "ice-shelf freshwater" upon its first usage in the abstract.

**Reviewer 2**

This is an interesting and generally well-written paper that discusses the drivers of ice shelf melt regime change in the south-western Weddell Sea as inferred from a relatively coarse resolution global ocean/sea ice model. The authors discuss how modifications to the model can reduce present day biases in the results, and that those changes are critical to the preservation of the current regime under current climate forcing. They further investigate the occurrence of a domino effect whereby increased ice shelf melting in the eastern Weddell Sea can trigger regime change in the south-west and describe how coarse resolution models may be pre-disposed to such a change.

The paper is logically structure and the results clearly presented. The text is pleasingly free of typographic errors and easy to read. Overall, I would recommend publication more-or-less as is, although the clarity could be improved further with a few minor modifications along the lines suggested below.

We thank the reviewer for their feedback and appreciation for the goals of the manuscript.

Teleconnection: I question the use of the term teleconnection. As used in the atmospheric context it refers to a rapid connection between two regions, typically established through a standing planetary wave pattern. If I understand the discussion here, the links are established slowly through the advection of anomalies by the mean circulation. I think teleconnection is a slightly misleading description of such a process.

We appreciate the reviewer pointing out this incorrect application of this term. We have replaced all instances with the more appropriate terms "remote influence" and "remote connection".

Water masses: There is a long and often confusing history of water mass names and acronyms, many of them regionally specific, that have appeared in the literature. While this paper does a reasonable job at navigating a way through that, I think some things could be clearer. DSW is a term that is not clearly defined in the literature or in this paper. It is apparently not just another term for HSSW (as used elsewhere?) because at one point it is stated that model DSW corresponds with observed LSSW. While I think the meanings are for the most part clear, I would encourage the authors to think about adopting the Whitworth et al (1998, Antarctic Research Series, vol 75) classification. They defined four water masses in the shelf/slope region, and while precise boundaries in T/S are region specific, nomenclature and origin/role are consistent in a circum-continental sense. In their classification, Shelf Water (SW) is the key water mass, defined as water that is denser than the regional variety of MCDW. Thus, presence or absence of SW is the key determinant of the transition in melt regime. Personally, I think Whitworth et al did the community a great service in proposing that simpler, more consistent and more intuitive circum-Antarctic water mass classification, which could clarify the sort of arguments made in this paper were it to be more widely adopted.

Thank you for bringing our attention to this classification, as the lack of consensus in the literature is a source of frustration to us as well. Since our model configurations show significant water mass biases (largely shifted to lower salinity space), the Whitworth et al. classification proves misleading, classifying almost all of the modeled water in this region as ASSW. In the text of the manuscript, we intend the water mass terms to refer to the relative role that the simulated water masses play in the dynamics despite these biases. For this purpose, we believe that DSW, AASW, mWDW and WDW are adequate and connect well with recent literature (which has also tended to use regionally specific terms). We would also emphasize that the water mass definitions shown in Figure 4 are not used in any quantitative analyses (e.g., Figure 7 uses the thermocline); we intend these boundaries as reference points for the reader. We have revised the manuscript to provide additional clarity around the use of the term DSW, which encompasses both HSSW and LSSW in our uses (though our model does not have HSSW on the Weddell shelf).

Model: The model naming is sometimes a little unclear. It is referred to throughout by the acronym ESM, but it is not the ESM used in Commeau at al. It does not have the interactive atmosphere, but are there any other distinctions? The various modifications made to the model parallel modifications to the full ESM, I think. That point only becomes apparent (to this reader at least) when Figure 12 is discussed. I think it would help if the model structure and experiments were set more clearly in the context of the Commeau et al work in section 2.

This manuscript describes the same ocean and sea-ice components from E3SM that were used by Comeau et al. (2020), with the only difference being a prescribed atmosphere instead of the interactive atmosphere component. Reviewer 1 had a similar concern about referring to our model as E3SM throughout the manuscript when being run without the atmosphere model disabled. We have adjusted the language throughout the manucsript to emphasize we are using a "global ocean sea-ice model" configuration from the components of E3SM.

Region: A location map with key features and all the ice shelves mentioned would really help orient readers, especially those less familiar with the region.

This suggestion was also made by the other reviewer and is a good one. We have added a location map with key features.

Minor comments:

Line 43-44: I'm not sure what feature you are referring to as the "Antarctic Coastal Countercurrent".

We have changed this to Antarctic Coastal Current.

Line 54-55: "… the iceberg melt term; …".

We have fixed this typo.

Line 148-153: I'm confused by what appear to be slightly contradictory statements about the prescribed melt rates. First you give specific values, but then you say that values are progressively halved relative to the maximum. Most, but not all quoted values fit that description, but is it superfluous anyway given that you quote specific numbers? But maybe it is the numbers that are obsolete (?), as they do not appear to coincide with lines in Figure 11 (at least two of them don't).

We agree this description was unnecessarily complicated and also inaccurate. We have corrected the list of prescribed melt rates to match those actually used (0.58, 1, 2, 4, 8, and 16 m/yr). We have deleted the sentence about progressively halving melt rates; we agree it is unnecessary.

Line 165: "… complete forcing cycle is complete". I know what you mean but the wording is a little odd. A complete cycle is complete by definition.

We updated this to say "full forcing cycle is complete".

Line 182: "… highest melt rates occur near the grounding …".

We have corrected the tense.

Line 233: "… on the Weddell Sea continental shelf (Fig. 6)."

We will correct the omission of "Sea". We will cite both Figure 6 and 7 and adjust the text to reference the increase in salinity and density of DSW.

Line 279: " … of these leading to a FRIS …"

We have corrected this typo.

Line 289: "… simulations experiencing rapid transition …".

We have corrected this typo.

Line 321:  This is now the first citation of Figure 7, if my earlier correction is OK.  I still don't really see that Figure 7 supports the statement.  Did you mean to refer to a different figure again?  Maybe figure 6 again?  In which case, is Figure 7 needed at all?

This statement was referring to small changes above the thermocline that are apparent in Fig. 7.  We have clarified the statement by adding the clause "namely, the increase in density above the thermocline due to the change in iceberg distribution is less than the 0.1 kg m^{-3}".

Line 414:  "… rapid change to continental shelf temperature …".

We have corrected this omission.

Figure 9 caption:  Note to self needs deleting.

We have deleted this wayward note.

---

## Author Response (AR2)

egusphere-2023-2226
Ice-shelf freshwater triggers for the Filchner-Ronne Ice Shelf melt tipping point in a global ocean-sea ice model

Matthew J. Hoffman, Carolyn Branecky Begeman, Xylar S. Asay-Davis, Darin Comeau, Alice Barthel, Stephen F. Price, and Jonathan D. Wolfe

Response to Editor
April 16, 2024

L20: is the greatest uncertainty in → is the most uncertain contributor to
*Text changed as suggested*

L248: also is → is also
*Text changed as suggested*

L275: , improving the → and improving the
*Text changed to ", which improves"*